# MiRNAs in Alcohol-Related Liver Diseases and Hepatocellular Carcinoma: A Step toward New Therapeutic Approaches?

**DOI:** 10.3390/cancers15235557

**Published:** 2023-11-23

**Authors:** Mickaël Jouve, Rodolphe Carpentier, Sarra Kraiem, Noémie Legrand, Cyril Sobolewski

**Affiliations:** Univ. Lille, Inserm, CHU Lille, U1286-INFINITE-Institute for Translational Research in Inflammation, F-59000 Lille, France; mickael.jouve@univ-lille.fr (M.J.); sarra.kraiem@inserm.fr (S.K.); noemie.legrand@univ-lille.fr (N.L.)

**Keywords:** microRNAs, alcohol-related liver disease, hepatocellular carcinoma, miRNAs-based therapeutics

## Abstract

**Simple Summary:**

Alcohol-related Liver Disease (ALD) is the leading cause of chronic liver disorders and the first cause of hepatocellular carcinoma in developed countries. Unfortunately, few and poorly efficient therapeutic options are available. Deciphering the molecular mechanisms underlying the development of these diseases is therefore of major interest. MicroRNAs (miRNAs) represent key regulators of gene expression by promoting mRNA decay and/or translation inhibition. Due to their ability to control the expression of many genes involved in metabolism, fibrosis, inflammation, and hepatic carcinogenesis, miRNAs represent potential therapeutic targets. Herein, we discuss the role of miRNAs in the different stages of ALD and their role in the onset of HCC, as well as the potential therapeutic options that could be envisaged.

**Abstract:**

Alcohol-related Liver Disease (ALD) is the primary cause of chronic liver disorders and hepatocellular carcinoma (HCC) development in developed countries and thus represents a major public health concern. Unfortunately, few therapeutic options are available for ALD and HCC, except liver transplantation or tumor resection for HCC. Deciphering the molecular mechanisms underlying the development of these diseases is therefore of major importance to identify early biomarkers and to design efficient therapeutic options. Increasing evidence indicate that epigenetic alterations play a central role in the development of ALD and HCC. Among them, microRNA importantly contribute to the development of this disease by controlling the expression of several genes involved in hepatic metabolism, inflammation, fibrosis, and carcinogenesis at the post-transcriptional level. In this review, we discuss the current knowledge about miRNAs’ functions in the different stages of ALD and their role in the progression toward carcinogenesis. We highlight that each stage of ALD is associated with deregulated miRNAs involved in hepatic carcinogenesis, and thus represent HCC-priming miRNAs. By using in silico approaches, we have uncovered new miRNAs potentially involved in HCC. Finally, we discuss the therapeutic potential of targeting miRNAs for the treatment of these diseases.

## 1. Introduction

Fatty Liver Diseases encompasses a spectrum of liver alterations associated with viral infection (e.g., hepatitis C), obesity, type 2 diabetes (Non-Alcoholic Fatty Liver Disease), and chronic/abusive alcohol consumption (Alcoholic Liver Disease) [1,2,3]. FLD starts with the development of hepatic steatosis, where hepatocytes accumulate lipids (i.e., triglycerides and cholesterol esters) [4,5]. With time, this step promotes chronic inflammation (steatohepatitis), which together with other defects (e.g., lipotoxicity, oxidative stress, endoplasmic reticulum (ER) stress, mitochondrial dysfunctions) trigger hepatocyte death [6,7]. In this context, fibrosis can develop and progress towards cirrhosis [8,9], a major cause of mortality and a high-risk condition for hepatocarcinogenesis [10]. Moreover, acute hepatitis (AH) can occur in patients with ALD, which is associated with severe liver failure and a high short-term mortality [11]. Hepatocellular carcinoma (HCC) represents the seventh most common cancer worldwide and the fourth most common cause of cancer mortality in both genders (https://gco.iarc.fr/ accessed on 3 October 2023). Because ALD is one of the most prevalent causes of chronic liver disease in developed countries, it is currently estimated that one-third of HCC develops in the context of alcoholic cirrhosis worldwide, with a strong heterogeneity between countries [12,13,14,15]. Moreover, the incidence of HCC is expected to dramatically increase in the future given the high prevalence of ALD in developed countries and the rapid worldwide increase in other risk factors, such as obesity/diabetes, which synergize with alcohol [16,17,18]. The prevalence of alcohol-associated cirrhosis was estimated at 0.3% in general populations [19]. Therefore, ALD is a major public health concern and a growing economic burden. Unfortunately, few therapeutic options are available for AH, such as corticoids, but this approach is strongly limited by the development of resistance [20,21,22]. HCC is also a poorly curable cancer, highly resistant to conventional chemotherapy and radiotherapy. To date, the most efficient treatment for advanced fibrosis, cirrhosis, and HCC remains liver transplantation [23]. Deciphering the molecular mechanisms underlying ALD and HCC development is therefore urgently needed to design new and effective therapeutic approaches.

Trans-acting factors controlling the fate of mRNAs (degradation, translation), such as microRNAs, are of high interest, due to their capacity to control the expression of a wide range of genes involved in various physiological and pathological processes (e.g., lipid, glucose metabolism, inflammation, fibrosis, and cancer-related processes). Accordingly, alteration of miRNA expression or activity contributes to the development of several diseases [24,25,26,27]. Although intense efforts have been devoted to characterizing miRNA functions in the context of NAFLD [28,29], a limited amount of knowledge is available for ALD and ALD-associated HCC. The purpose of this review is to discuss the role of microRNAs in the different stages of ALD and how they contribute to the progression toward HCC (HCC priming events). Finally, we discuss the different strategies that could be employed to target miRNAs in ALD and ALD-related HCC.

## 2. MicroRNAs

MicroRNAs are small endogenous non-coding RNAs of 16–22 nucleotides, controlling gene expression at the post-transcriptional level by recognizing complementary sequences within the 3′ Untranslated Region (UTR) of targeted mRNAs and promoting either mRNA decay and/or translation inhibition [30,31]. Since their discovery in 1993 (lin4 in C-Elegans) [32], more than 38,589 miRNAs (miRBase) have been identified in different organisms (i.e., plants, animals, viruses), among which many have been associated to a wide range of physiological and pathological processes [24,25,26,27].

MiRNA biogenesis encompasses several steps, starting with the transcription of a primary miRNA transcript (pri-miRNA) from intronic or intergenic regions by the RNA polymerase II/III [33]. The pri-miRNA is processed in 3′ and 5′ strands by the microprocessor complex comprising the ribonuclease III enzyme, Drosha, and the RNA-Binding Protein (RBP) DiGeorge Syndrome Critical Region 8 (DGCR8), thereby generating a precursor miRNA (pre-miRNA). The pre-miRNA is exported from the nucleus to the cytosol by the Exportin5/RanGTP. In the cytosol, the pre-miRNA is processed by the RNase III endonuclease Dicer, which removes the terminal loop of the pri-miRNA thereby producing a mature miRNA duplex, composed of a guide strand and a complementary passenger strand [34]. According to the canonical model, the passenger strand is degraded, while the guide strand is maintained and is incorporated into the RNA-Induced Silencing Complex (RISC) to settle at the complementary sequences in the 3′UTRs of their target’s mRNAs. However, this dogmatic view is currently challenged by several physiological/pathological situations where the passenger strand is conserved and also exerts important regulatory functions.

MiRNA-dependent regulation is a complex regulator process as evidenced by their capacity to control the expression of a wide range of transcripts. Conversely, one mRNA can be regulated by several miRNAs [30,35]. Moreover, our understanding of miRNAs-dependent regulation is challenged by the interplay between miRNAs, long non-coding RNA (lncRNAs), and RNA Binding Proteins (RBPs), which importantly control the expression, but also the bioavailability and activity of miRNAs [34]. While most studies are focusing on miRNAs with a deregulated expression pattern, increasing evidence indicates that the expression does not always correlate with the activity of miRNAs. Deciphering this interplay is therefore important to identify the most relevant and active miRNAs involved in pathological contexts and thus to design efficient therapeutic approaches.

This review summarizes the role of miRNAs in the different steps of ALD and discuss how these alterations promote hepatic carcinogenesis (Figure 1).

**Figure 1 cancers-15-05557-f001:**
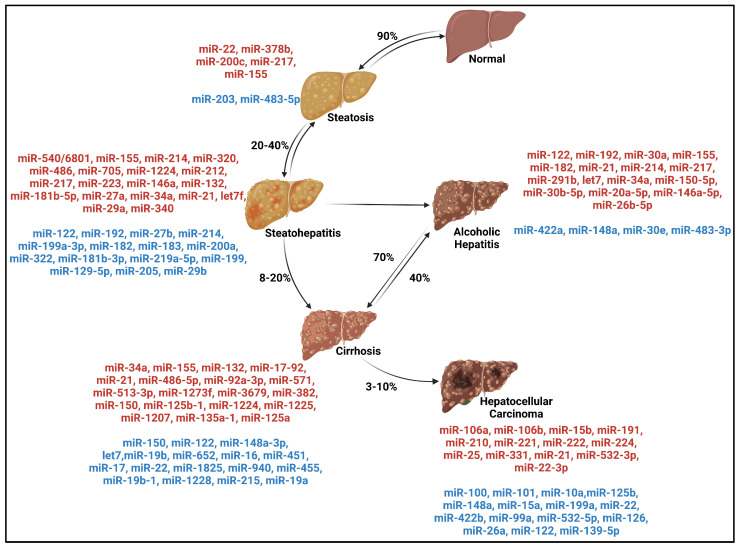
Spectrum of Alcohol-related Liver Disease with deregulated microRNAs at each stage and cited in this review and detailed in Table 1. Percentages represent the rate of patients moving from one stage to another [36]. MicroRNAs in red have increased expression and microRNAs in blue have decreased expression. Created by Biorender.com.

**Table 1 cancers-15-05557-t001:** Summary of deregulated miRNAs in ALD and cited in this review.

MiRs	Expression	Function	Target	Model	Cell Types	Refs.
**Alcohol-Related Steatosis**						
MiR-203	Down	Decrease lipid accumulation	Lipin1	AML12	Hepatocyte	[37]
MiR-483-5p	Down	Steatosis cell proliferation	PPARα	Human Mice HepaRG	Hepatocyte	[38]
MiR-22	Up	Steatosis	FGFR1 FGF21 IL6/JAK/STAT	Human Mice	Hepatocyte	[39,40,41]
MiR-378b	Up	Lipid accumulation	CAMKK2	Mice Human Hepatocyte	Hepatocyte	[42]
MiR-200c	Up	Modulation of lipid homeostasis	Hnf1 Homeobox B	Mice	Liver	[43]
MiR-217	Up	Inflammation Steatosis	Sirtuin-1	Mice RAW 264.7 Kupffer cells	Hepatocyte Macrophage Kupffer cells	[44,45]
MiR-155	Up	Promote liver steatosis Liver injury Inflammation Fibrosis	Snail1 Smad2 STAT3 PPARα TLR inhibitor PPARγ TNFα	Human Mice Raw 264.7 Hepa 1-6	Kupffer cells Hepatocyte Hepatic stellate cells	[46,47,48,49,50]
**Alcoholic Steatohepatitis**						
MiR-122	Down	Protection against steatosis fibrosis	HiF1α TNFrsf13C	Human Mice RAW 264.7 Huh7	Hepatocyte Kupffer cells Extracellular vesicles	[51,52,53,54]
MiR-192	Down	Exosome induction	Rab27a Rab35 STX7 STX16	Human	Hepatocyte Extracellular vesicles	[53,55]
MiR-27b	Down	Inflammation	LPS	Mice RAW 264.7	Macrophage	[56]
MiR-214	Down (mice) Up (human)	Liver fibrogenesis Induce oxidative stress	Gluthatione reductase	Human Mice Rat Bel7402 BRL	Hepatocyte	[40,57,58]
MiR-199a-3p	Down	n.a	n.a	Mice	n.a	[57]
MiR-182	Down	Inflammation Apoptosis	Mcp-1 Ccl20 Cxcl5 Cxcl1 Bcl2	Mice Human	Liver	[40,57]
MiR-183	Down	Inflammatory	n.a	Mice	n.a	[57]
MiR-200a	Down	Disease severity	Gli2	Mice	n.a	[57,59,60]
MiR-322	Down	n.a	n.a	Mice	n.a	[57]
MiR-181b-3p	Down	Inflammatory	Importin α5	Mice Rat	Kupffer cells	[61,62]
MiR-219a-5p	Down	Oxidative stress	P66shc	Rat AML-12	Hepatocyte	[63]
MiR-199	Down	Inflammation	Hif1α	Rat	Kupffer cells	[64]
MiR-129-5p	Down	Hepatic fibrosis Lipid metabolism	NEAT1	Human Mice AML-12	Hepatocyte	[65]
MiR-540	Up	Hepatic steatosis Oxidative stress	PPARα ACOX1	Mice	n.a	[66]
MiR-6801	Up	Hepatic steatosis Oxidative stress	PPARα ACOX1	Human	n.a	[66]
MiR-155	Up	Promote liver steatosis Liver injury Inflammation fibrosis	Snail1 Smad2 STAT3 PPARα TLR inhibitor PPARγ TNFα	Human Mice Raw 264.7 Hepa 1-6	Kupffer cells Hepatocyte Hepatic stellate cells	[46,47,48,49,50]
MiR-320	Up	Inflammatory	n.a	Mice	n.a	[57]
MiR-486	Up	Inflammatory	n.a	Mice	n.a	[57]
MiR-705	Up	Inflammatory	n.a	Mice	n.a	[57]
MiR-1224	Up	Inflammatory Tumor suppressor	n.a	Mice Human	Liver	[40,57]
MiR-212	Up	Gut leakiness	ZO-1	Mice	Gut epithelial cells	[67,68]
MiR-223	Up	Inflammation Liver injury	IL-6 p47^phox^ NFκB	Human Mice	Neutrophils Kupffer cells	[69,70,71]
MiR-146a	Up	Anti-inflammatory	TLR	Human Mice	Monocyte Kupffer cells	[72]
MiR-132	Up	Inflammation Fibrosis	αSMA Collagen fibers Caspase 3 extracellular vesicles	Human Mice	Kupffer cells Hepatic stellate cells	[73,74]
MiR-181b-5p	Up	Oxidative stress Inflammation	PIAS1	Rat	Hepatocyte	[75]
MiR-27a	Up	Fibrosis monocyte differentiation	ERK Sprouty 2 Nr1d2 CD206	HumanHuh7.5 cells Monocytes	Kuppfer cells Monocytes Extracellular vesicles	[76,77,78]
MiR-34a	Up	Fibrosis Cellular senescence Mallory–Denk cell formation	Smad3 SIRT1	Human Mice	Kuppfer cells Hepatocyte Hepatic stellate cells Mallory–Denk cells	[79,80,81,82,83]
MiR-21	Up	Regulate hepatic cell survival, transformation, and remodel liver regeneration	VHL Fas ligand (TNF superfamily, member 6) (FASLG) and death receptor 5 (DR5)	Rat Human Mice	Hepatic stellate cells Kuppfer cells Hepatocyte	[40,84,85,86]
Let-7f	Up	Potential biomarkers Potential mediators of intercellular crossovers	n.a	Mice	Extracellular vesicles	[87]
MiR-29a	Up	Potential biomarkers Potential mediators of intercellular crossovers	n.a	Mice	Extracellular vesicles	[87]
MiR-340	Up	Potential biomarkers Potential mediators of intercellular crossovers	n.a	Mice	Extracellular vesicles	[87]
MiR-205	Down	Inflammation	Importinα5	Mice	Kupffer cells	[88]
MiR-29b	Down	Inflammation	STAT3	Mice RAW264.7	Kupffer cells	[89]
MiR-217	Up	Inflammation Steatosis	Sirtuin-1	Mice RAW 264.7 Kupffer cells	Hepatocyte Macrophage Kupffer cells	[44,45]
**Cirrhosis**						
MiR-150	Up Down	Antifibrotic Tumor suppressor	αSMA Col1A1	Human	Hepatic stellate cells	[40,90]
MiR-148a-3p	Down	Fibrosis	ERBB3	Rat	Hepatic stellate cells	[91]
Let-7	Down	Fibrosis Inflammatory	Lin28 TLR7	Mice Human	Hepatic stellate cells	[92,93]
MiR-19b	Down	HSCs activation	Pri-miR-17-92 TGFβRII MeCP2	Rat LX2 HepG2	Hepatic stellate cells	[94]
MiR-652	Down	n.a	n.a	Human	n.a	[95,96]
MiR-16	Down	n.a	n.a	Human	Exosome	[97]
MiR-451	Down	Tumor suppressor	n.a	Human	Liver	[40]
MiR-17	Down	Tumor suppressor	n.a	Human	Liver	[40]
MiR-1825	Down	Tumor suppressor	n.a	Human	Liver	[40]
MiR-940	Down	Tumor suppressor	n.a	Human	Liver	[40]
MiR-455	Down	Tumor suppressor	n.a	Human	Liver	[40]
MiR-19b-1	Down	Tumor suppressor	n.a	Human	Liver	[40]
MiR-1228	Down	OncomiR	n.a	Human	Liver	[40]
MiR-215	Down	OncomiR	n.a	Human	Liver	[40]
MiR-19a	Down	OncomiR	n.a	Human	Liver	[40]
MiR-17-92	Up	Fibrogenesis	n.a	n.a	Hepatic stellate cells	[94]
MiR-486-5p	Up	n.a	n.a	Human	n.a	[95,96]
MiR-92a-3p	Up	n.a	n.a	Human	n.a	[95,96]
MiR-571	Up	n.a	CREBBP	Human	Hepatic stellate cells	[95,96,98]
MiR-513-3p	Up	n.a	n.a	Human	n.a	[95,96]
MiR-1273f	Up	OncomiR	n.a	Human	Liver	[40]
MiR-3679	Up	OncomiR	n.a	Human	Liver	[40]
MiR-382	Up	OncomiR	n.a	Human	Liver	[40]
MiR-125b-1	Up	Tumor suppressor	n.a	Human	Liver	[40]
MiR-1225	Up	Tumor suppressor	n.a	Human	Liver	[40]
MiR-1207	Up	Tumor suppressor	n.a	Human	Liver	[40]
MiR-135a-1	Up	Tumor suppressor	n.a	Human	Liver	[40]
MiR-125a	Up	Tumor suppressor	n.a	Human	Liver	[40]
MiR-22	Down	Steatosis Tumor suppressor Deregulated pathways in HCC	FGFR1 FGF21 IL6/JAK/STAT	Human Mice	Hepatocyte	[39,40,41]
MiR-122	Down	Protection against steatosis fibrosis	HiF1α TNFrsf13C	Human Mice RAW 264.7 Huh7	Hepatocyte Kupffer cells Extracellular vesicles	[51,52,53,54]
MiR-155	Up	Promote liver steatosis Liver injury Inflammation fibrosis	Snail1 Smad2 STAT3 PPARα TLR inhibitor PPARγ TNFα	Human Mice Raw 264.7 Hepa 1-6	Kupffer cells Hepatocyte Hepatic stellate cells	[46,47,48,49,50]
MiR-1224	Up	Inflammatory Tumor suppressor	n.a	Mice Human	n.a	[40,57]
MiR-132	Up	Inflammation Fibrosis	αSMA Collagen fibers Caspase 3 extracellular vesicles	Human Mice	Kupffer cells Hepatic stellate cells	[73,74]
MiR-34a	Up	Fibrosis Cellular senescence Mallory–Denk cell formation	Smad3 SIRT1	Human Mice	Kuppfer cells Hepatocyte Hepatic stellate cells Mallory–Denk cells	[79,80,81,82,83]
MiR-21	Up	Regulate hepatic cell survival, transformation, and remodel liver regeneration	VHL Fas ligand (TNF superfamily, member 6) (FASLG) and death receptor 5 (DR5)	Rat Human Mice	Hepatic stellate cells Kuppfer cells Hepatocyte	[40,84,85,86]
**Alcoholic Hepatitis**						
MiR-422a	Down	n.a	n.a	Human	n.a	[40]
MiR-30b-5p	Up	Associated mortality	n.a	Human	Extracellular vesicles	[99]
MiR-20a-5p	Up	Associated mortality	n.a	Human	Extracellular vesicles	[99]
MiR-26b-5p	Up	Associated mortality	n.a	Human	Extracellular vesicles	[99]
MiR-148a	Down	Anti-inflammatory Deregulated pathways in HCC	TXNIP Epigenetics TGFβ PI3K/AKT	Human Mice	Hepatocyte	[41,100]
MiR-30e	Down	Inflammation	UCP2 ATP H_2_O_2_	Mice	n.a	[101]
MiR-483-3p	Down	Mallory–Denk cell formation	BRCA1	Human	Mallory–Denk cells	[82]
MiR-146a-5p	Up	Associated mortality	n.a	Human	Extracellular vesicles	[99]
MiR-30a	Up	Autophagy	Beclin-1	Human	Exosome	[53,102]
MiR-291b	Up	Inflammation	Tollip	Human Rat	Kupffer cells	[103]
MiR-150-5p	Up	Cell death	CISH	Human	Liver	[104]
MiR-217	Up	Inflammation Steatosis	Sirtuin-1	Mice RAW 264.7 Kupffer cells	Hepatocyte Macrophage Kupffer cells	[44,45]
MiR-122	Up	Protection against steatosis fibrosis	HiF1α TNFrsf13C	Human Mice RAW 264.7 Huh7	Hepatocyte Kupffer cells Extracellular vesicles	[51,52,53,54]
MiR-192	Up	Exosome induction	Rab27a Rab35 STX7 STX16	Human	Hepatocyte Extracellular vesicles	[53,55]
MiR-214	Up	Liver fibrogenesis Induce oxidative stress	Gluthatione reductase	Human Mice Rat Bel7402 BRL	Hepatocyte	[40,57,58]
MiR-182	Up	Inflammation Apoptosis	Mcp-1 Ccl20 Cxcl5 Cxcl1 Bcl2	Mice Human	Liver	[40,57]
MiR-155	Up	Promote liver steatosis Liver injury Inflammation fibrosis	Snail1 Smad2 STAT3 PPARα TLR inhibitor PPARγ TNFα	Human Mice Raw 264.7 Hepa 1-6	Kupffer cells Hepatocyte Hepatic stellate cells	[46,47,48,49,50]
MiR-34a	Up	Fibrosis Cellular senescence Mallory–Denk cells formation	Smad3 SIRT1	Human Mice	Kuppfer cells Hepatocyte Hepatic stellate cells Mallory–Denk cells	[79,80,81,82,83]
MiR-21	Up	Regulates hepatic cell survival, transformation, and remodeling Liver regeneration	VHL Fas ligand (TNF superfamily, member 6) (FASLG) and death receptor 5 (DR5)	Rat Human Mice	Hepatic stellate cells Kuppfer cells Hepatocyte	[40,84,85,86]
Let-7	Up	Fibrosis Inflammatory	Lin28 TLR7	Mice Human	Hepatic stellate cells	[92,93]
**Hepatocellular carcinoma**						
MiR-100	Down	Deregulated pathways in HCC	IGF signaling	Human	Liver	[41]
MiR-101	Down	Deregulated pathways in HCC	Epigenetics TGFβ PI3K/AKT TP53/Cell cycle	Human	Liver	[41]
MiR-10a	Down	Deregulated pathways in HCC	MAPK Wnt/βCat	Human	Liver	[41]
MiR-125b	Down	Deregulated pathways in HCC	TP53/Cell cycle IL6/JAK/STAT IGF signaling	Human	Liver	[41]
MiR-15a	Down	Deregulated pathways in HCC	TGFβ	Human	Liver	[41]
MiR-199a	Down	Deregulated pathways in HCC	TGFβ	Human	Liver	[41]
MiR-422b	Down	Deregulated pathways in HCC		Human	Liver	[41]
MiR-99a	Down	Deregulated pathways in HCC	IGF signaling	Human	Liver	[41]
MiR-139-5p	Down	Deregulated pathways in HCC		Human	Liver	[41]
MiR-106a	Up	Deregulated pathways in HCC	Epigenetics	Human	Liver	[41]
MiR-106b	Up	Deregulated pathways in HCC	Epigenetics TGFβ	Human	Liver	[41]
MiR-15b	Up	Deregulated pathways in HCC	TP53/Cell cycle	Human	Liver	[41]
MiR-191	Up	Deregulated pathways in HCC	Wnt/βCat NFκB TP53/Cell cycle	Human	Liver	[41]
MiR-210	Up	Deregulated pathways in HCC	n.a	Human	Liver	[41]
MiR-221	Up	Deregulated pathways in HCC	PI3K/AKT TP53/Cell cycle	Human	Liver	[41]
MiR-222	Up	Deregulated pathways in HCC	Wnt/βCat PI3K/AKT	Human	Liver	[41]
MiR-224	Up	Deregulated pathways in HCC	PI3K/AKT TP53/Cell cycle IL6/JAK/STAT	Human	Liver	[41]
MiR-25	Up	Deregulated pathways in HCC	Wnt/βCat	Human	Liver	[41]
MiR-331	Up	Deregulated pathways in HCC	n.a	Human	Liver	[41]
MiR-532-3p	Up	Promotes HCC cells migration, invasion, and proliferation	Protein tyrosine phosphatase receptor type T (PTPRT)	HCC specimens Hep3B HepG2 SMMC-7721 Huh7 MHCC-97 H	Hepatocyte	[105]
MiR-532-5p	Down	Promotes cell proliferation and metastasis	Chemokine (C-X-C motif) ligand 2 (CXCL2), X-ray Repair Cross Complementing 5 (XRCC5)	HEL7702 HEL7404 HCCLM3 SMMC7721 HepG2 PG5 MHCC97H Huh7	Hepatocyte	[106,107]
MiR-22-3p	Up	Promotes HCC cells’ stemness and metastasis	Ten-eleven-translocation 2 (TET2)	HCC specimens Xenograft on BALB/C nude mice HCCLM3	Cancer stem cells	[108]
MiR-126	Down	Suppresses cell proliferation, invasion and migration	Epithelial Growth Factor Receptor (EGFR)	HCC specimens Hep3B MHCC97H Huh7 HCCLM3	Hepatocyte Cancer stem cells	[109]
MiR-26a	Down	n.a	n.a	HCC specimens	n.a	[110]
MiR-22	Down	Steatosis Tumor suppressor Deregulated pathways in HCC	FGFR1 FGF21 IL6/JAK/STAT	Human Mice	Hepatocyte	[39,40,41]
MiR-122	Down	Protection against steatosis fibrosis	HiF1α TNFrsf13C	Human Mice RAW 264.7 Huh7	Hepatocyte Kupffer cells Extracellular vesicles	[51,52,53,54]
MiR-21	Up	Regulate hepatic cell survival, transformation, and remodel liver regeneration	VHL Fas ligand (TNF superfamily, member 6) (FASLG) and death receptor 5 (DR5)	Rat Human Mice	Hepatic stellate cells Kuppfer cells Hepatocyte	[40,84,85,86]
MiR-148a	Down	Anti-inflammatory Deregulated pathways in HCC	TXNIP Epigenetics TGFβ PI3K/AKT	Human Mice	Hepatocyte	[41,100]

## 3. MicroRNAs in Alcohol-Induced Steatosis

ALD starts with the accumulation of lipids (i.e., triglycerides) in hepatocytes (steatosis). This effect is associated with the metabolism of alcohol in hepatocytes and the impact of ethanol on adipocytes [39]. The metabolic pathways governing alcohol-induced steatosis (e.g., AMPK, PPARα, SREBP-1) are finely tuned by miRNAs, which directly and indirectly control the expression of key enzymes of lipid metabolism. While some miRNAs contribute to hepatic steatosis, others are deregulated as a compensatory mechanism to overcome the excess of lipid storage. Targeting pro-lipogenic miRNAs or, in contrast, restoring the expression of “protective/gate keeper” miRNAs are therefore of high interest for therapeutic purposes (Figure 2 and Table 1). The main metabolic processes involved in alcohol-induced steatosis under miRNA dependency are discussed below.

### 3.1. Ethanol Metabolism

The liver metabolizes 90–95% of blood ethanol by the concerted action of several metabolizing enzymes. First, alcohol is metabolized into acetaldehyde by three pathways involving the cytosolic alcohol dehydrogenase (ADH), peroxisomal catalase, or microsomal cytochrome P450 2E1 (CYP2E1). Acetaldehyde is then detoxified into acetate by the mitochondrial aldehyde dehydrogenase 2 (ALDH2) in an NAD^+^/NADH-dependent manner [111]. In the case of chronic and excessive alcohol consumption, the ethanol-inducible CYP2E1 pathway feeds the oxidative phosphorylation, thereby enhancing oxidative stress in hepatocytes [112]. Acetaldehyde exerts pleiotropic effects to promote fat accumulation in hepatocytes (Figure 2). First, acetaldehyde reduces peroxisome proliferator-activated receptor α (PPARα) activity, thereby decreasing β-oxidation. Second, acetaldehyde reduces AMP-activated protein kinase (AMPK) activity, and thus increases the activity of acetyl-coA carboxylase (ACC) [113]. Finally, acetaldehyde increases the expression of the transcription factor sterol regulatory element binding protein 1 (*SREBP-1*), which controls the expression of several lipogenic enzymes (e.g., fatty acid synthase, *FASN*). Several miRNAs have been involved in the regulation of ADH, CYP2E1, or ALDH and thus are important regulators of alcohol-induced hepatic steatosis. For instance, miR-214-3p and miR-552 directly regulate CYP2E1 expression, as evidenced in hepatic cancer cells (HepG2) [114]. Interestingly, miR-552 inhibits the transcription of CYPE21, through its capacity to bind to the promoter region and prevents the binding of SMARCE1 (SWI/SNF-Related, Matrix-Associated, Actin-Dependent Regulator Of Chromatin, Subfamily E, Member 1) and RNA polymerase II [115]. This later illustrates well the importance of “non-canonical” mechanisms of miRNA-dependent gene expression regulation.

### 3.2. FGF21 and AMPKα Signaling

An increase in miR-22 expression has been observed in fatty livers from mice fed a Lieber–DeCarli (LDC) diet and from patients with a history of alcohol consumption [39]. MiR-22 directly inhibits FGF21 expression, inhibiting PPARα and PGC1α binding in its regulatory region, as well as its FGFR1 receptor in hepatocytes, thereby reducing AMPKα activity and increasing hepatic lipogenesis [39]. The activity of AMPK is also indirectly reduced by alcohol-induced miR-378b in human hepatocytes and in ethanol-fed mice [42]. Indeed, miR-378b directly targets the Ca^2+^/calmodulin-dependent protein kinase kinase 2 (CaMKK2), a positive regulator of AMPK [42].

### 3.3. PPARα/γ Signaling

Peroxisome proliferator-activated receptors (PPARs) play an important role in hepatic steatosis. While PPARα promotes β-oxidation and inhibits triglyceride biosynthesis, PPARγ activity is, in contrast, upregulated following ethanol exposure, thus activating SREBP-1c and its downstream target genes involved in lipogenesis (e.g., *FASN*, *DGAT1*, *DGAT2*) [116]. PPARα is downregulated by acetaldehyde during alcohol consumption [113]. MiR-155, which is induced in the liver of alcohol-fed mice, importantly contributes to hepatic steatosis by directly inhibiting PPARα expression [47]. Interestingly, some miRNAs may control PPARα indirectly, as suggested for miR-203, which is downregulated in the liver of mice fed a Gao-Binge alcoholic diet (an alcohol-enriched diet coupled with a single binge ethanol administration). MiR-203 directly upregulates LPIN1 (Lipin-1), a transcriptional co-activator of PPARα [37]. Paradoxically, miR-483-5p, a direct regulator of PPARα is downregulated in alcohol-fed mice (Lieber–DeCarli diet), thus suggesting a protective mechanism aiming at lowering intracellular lipid content [38]. Finally, although PPAR*γ* expression is highly regulated by miRNAs in the liver [117], there is currently no evidence of this link in the context of ALD.

### 3.4. SREBP Signaling

SREBP is a major transcription factor transactivating lipogenesis-related genes (e.g., *FASN, ACACA*) [118], and its regulation by microRNAs has been extensively documented in the context of NAFLD [119,120]. As described above, the downregulation of miR-203 expression in the liver of mice fed an alcoholic diet [37], is directly responsible for the upregulation of Lipin-1 [37], which can promote β-oxidation and inhibit SREBP-1 signaling. In addition, Lipin-1 acts as a Mg2+-dependent phosphatidate phosphatase (PAP) enzyme involved in phospholipid and triacylglycerol (TAG) biosynthesis depending on its localization [121,122]. Alcohol-induced miR-217 and mir-200c overexpression also contribute to the activation of SREBP by downregulating the expression of *SIRT1* and *HNF1B* in hepatocytes [43,44,123,124]. Finally, other factors involved in the maturation of SREBP-1 [125], such as early growth response-1 (EGR1), which is activated by alcohol consumption, are regulated by miRNAs [126,127,128]. However, this regulation remains poorly known in the context of ALD [129].

### 3.5. Lipolysis in the Adipose Tissue

Ethanol induces hepatic steatosis indirectly by promoting lipolysis in the adipose tissue, thereby releasing free fatty acids (FFAs), which are imported by the liver by specific transporters (e.g., CD36) [130]. The regulation of CD36 by miRNAs in the context of ALD is currently unknown but several miRNAs have been uncovered in other hepatic diseases (Non-Alcoholic Fatty Liver Disease), such as miR-29a [131], miR-20a-5p [132], or miR-26a [133]. Furthermore, alcohol-induced hepatic steatosis has been associated with the release of FGF21 (Fibroblast Growth Factor 21) in the plasma. FGF21 triggers a systemic elevation of catecholamine by the sympathetic nervous system, which binds to the β-adrenergic receptor on adipocytes, raising intracellular cAMP and activating lipolytic enzymes [134].

### 3.6. Alcohol-Related Steatosis as a Priming Event for Hepatocarcinogenesis?

Several deregulated miRNAs in alcohol-induced hepatic steatosis have previously been associated with HCC development, thus suggesting that these early alterations pave the way for hepatic carcinogenesis. Indeed, downregulation of the miR-200 family promotes cancer progression and development [135]. The downregulation of miR-483-5p activates NOTCH3 signaling [38], a pro-tumorigenic pathway involved in HCC and associated with a poor prognosis [136]. Other miRNAs such as miR-203 or miR-22 are downregulated with steatosis and exert tumor-suppressive functions. Indeed, miR-203 inhibits hepatic cancer cells proliferation and metastasis [137], and miR-22 directly inhibits cyclin A2 expression [138].

## 4. MicroRNAs in Alcoholic Steatohepatitis (ASH)

The chronic accumulation of lipids, together with other damages (e.g., oxidative stress, mitochondrial dysfunction, altered liver–gut axis) promotes a chronic and low-grade inflammation mediated by the innate immune system [139,140], which is commonly referred as Alcoholic Steatohepatitis (ASH). This step is also promoted by the high number of free radicals generated by the metabolism of ethanol, which triggers oxidative stress with lipid peroxidation, and important cellular damages [141]. Lipid peroxidation derivates, such as malondialehyde (MDA) and 4-hydroxy-2-nonenal (HNE), stimulate collagen production by HSCs. In HSCs, acetaldehyde modifies collagen carboxyl-terminal pro-peptide, thus affecting its capacity to exert a negative feedback control on collagen synthesis [142]. Acetaldehyde and MDA form hybrid adducts with proteins, known as malondialdehyde-acetaldehyde (MAA) adducts [143], which are recognized by Kupffer, endothelial, and stellate cells via scavenger receptors (e.g., CD36) and promote the production of pro-inflammatory cytokines (e.g., Il-1β, TNFα) and chemokines (e.g., MCP-1, MIP-2) [144]. MiRNAs play an important role in ASH by controlling several inflammatory pathways/processes. While some deregulated miRNAs favor ASH, others display anti-inflammatory properties. The development of ASH is therefore dependent on a disbalance between the detrimental and beneficial miRNAs. The most important pathways/processes underlying ASH and regulated by miRNAs are discussed below.

### 4.1. Altered Gut–Liver Axis Toll-like Receptor Signaling

By shifting the gut microbial composition towards pathogenic species (e.g., *Bacteroides* spp. *Stomatococcus*) [145], alcohol makes the intestinal epithelium more permeable to endotoxins and lipopolysaccharides (LPSs), which quickly reach the liver through the portal vein [146]. LPSs trigger inflammatory pathways (e.g., MyD88, NFκB signaling), through the activation of Toll-like receptors (TLRs) (eg., TLR4) expressed at the surface of Kupffer cells but also HSCs [147], and thus induce the expression of pro-inflammatory cytokines (e.g., TNFα, MCP-1) [148,149] and fibrogenic factors (e.g., TGFβ1) [150,151] (Figure 3). MiRNAs regulate the gut–liver axis, as evidenced for miR-212 in intestinal epithelial cells of alcohol-fed mice. Indeed, alcohol, through the action of acetaldehyde, increases inducible nitric oxide synthase (iNOS) signaling, leading to the overexpression of miR-212 in intestinal epithelial cells. MiR-212 inhibits the translation of Zonula occludens-1 (ZO-1), a major component of tight junctions involved in intestinal barrier permeability [67,68]. Alcohol- and gut-derived LPSs also trigger the overexpression of miR-217 in Kupffer cells and Raw 264.7 cells (mouse macrophages), which in turn directly inhibits the expression of sirtuin-1 (SIRT1). This effect promotes NFκB and nuclear factor of activated T-cells c4 (NFATc4) activities, and thus the expression of pro-inflammatory cytokines (i.e., TNFα and Il-6) [45]. MiR-155, which is upregulated with chronic alcohol consumption, inhibits negative regulators of TLR4 signaling (e.g., IRAK-M, SHIP1 and SOCS1), and thus promotes the Myd88/NFκB pathway and the increased level of TNFα [152]. MiR-181b-3p, which targets importin α5, is downregulated by alcohol and this effect promotes the expression of pro-inflammatory cytokines in Kupffer cells from ethanol-fed rats [61,62]. Finally, other miRNAs exerting anti-inflammatory properties have been reported in ASH, such as miR-146a, which decreases TLR signaling [72]. This miRNA is upregulated in ASH and thus may represent part of a defense/compensatory mechanism aiming at lowering overactivated pro-inflammatory signaling [153]. Potentiating the effect of anti-inflammatory miRNAs or, in contrast, inhibiting “inflammamiRs”, represent potential therapeutic approaches to consider.

### 4.2. PPARα/γ Signaling

The PPARα signaling importantly contributes to hepatic inflammation by inhibiting the NFκB pathway and the expression of associated pro-inflammatory cytokines [154]. In contrast, PPARγ promotes hepatic steatosis and reduces inflammation [155,156]. Therefore, miRNAs targeting PPARα or PPARγ are likely contributing to the development of ASH. In the liver of mice fed an LDC diet as well as in mouse primary hepatocytes, a decrease in ALR (Augmenter of Liver Regeneration) protein was observed and promoted hepatic steatosis and oxidative stress. This effect is mediated by the induction of miR-540 expression, which directly inhibits *Acox1* and *Pparα* expression. Although miR-540 is poorly conserved and not expressed in humans, miR-6801 has been identified as its functional equivalent. However, functional studies are still required to characterize the role of this miRNA in ASH [66]. MiR-122, which represents the most abundant miRNA in the liver (70% of the hepatic miRnome in adult mouse and 52% in human), also plays an important role in ASH development, as evidenced in miR-122 KO mice, which sequentially develop hepatic steatosis, inflammation, and hepatocellular carcinoma (HCC) [157]. Alcohol consumption induces an increase in the transcription factor granyhead-like transcription factor 2 (GRHL2) in murine hepatocytes, which inhibits the transcription of miR-122. In turn, miR-122 directly inhibits the expression of *Hif1α*, a factor that induces liver damage and increases the expression of PPARγ, a major component of lipogenesis [52]. Alcohol also decreases miR-192 expression in human hepatocytes [55]. This inhibition increases the expression of several targets of miR-192, including Rab27a, Rab35, syntaxin7 (STX7), and syntaxin16 (STX16), which are involved in extracellular vesicles [55]. MiR-155 is also an important miRNA involved in ASH, as evidenced by miR-155 KO mice, which are protected from alcohol-induced fat accumulation and inflammation. This effect has been associated with an increase in PPARα and a decrease in MCP-1 [48]. Together with miR-132, an increase in miR-155 expression was observed in LDC-fed mice (Figure 3) [73]. MiR-155 directly decreases PPARα in hepatocytes, thus promoting hepatic steatosis [48,49] (Figure 3). In Kupffer cells, miR-155, which is activated by the NFκB pathway, induces TNFα production [46,158] and also inhibits the expression of PPARγ [47], an inhibitor of the NFκB pathway [159].

### 4.3. NFκB Signaling

NFκB signaling is a major pathway promoting the expression of pro-inflammatory cytokines (e.g., TNFα, IL-1β) and mediators (e.g., COX-2). In the liver, NFκB is activated by different stimuli, such as LPSs, through the TLR/MyD88 pathway, or pro-inflammatory cytokines, such as TNFα or IL1-β [160]. The post-transcriptional regulation of NFκB signaling has been extensively documented in several disorders, including chronic liver diseases and HCC [161]. However, very few studies have depicted this regulation in the context of alcohol. Some miRs have been shown to inhibit NFκB signaling, such as miR-27b [56] and miR-223 [69], which are respectively down- and upregulated in the liver of LDC-fed mice [57]. MiR-205, which is downregulated in ALD, represses the NFκB pathway in ethanol-fed mouse Kupffer cells [88]. MiR-205 inhibits directly importinα5, a protein involved in nuclear transfer of the NFκB signaling pathway [162]. In macrophages (RAW 264.7 and Kupffer cells from alcohol-fed mice), the NFκB pathway activated by chronic alcohol exposure and LPS stimulation induces the expression of miR-155, which in turn increases TNFα production by increasing the stability of its mRNA [46]. MiR-217, which is upregulated in Kupffer cells from alcohol-fed mice, inhibits sirtuin1, an inhibitor of the NFκB pathway [45]. Finally, ethanol and LPS will also induce circ_1639 expression, a circular RNA activating the NFκB pathway and TNF Receptor Superfamily Member 13C (TNFrsf13C) gene expression by inhibiting miR-122 expression in macrophages (RAW 264.7 and Kupffer cells from alcohol-fed mice) [51].

### 4.4. Il-6/STAT3 Signaling

The IL-6/STAT3 pathway is a major component of chronic liver diseases and HCC development by controlling the innate immune response [163] but also the expression of mitogenic/survival factors in hepatocytes (e.g., MYC), thereby promoting hepatic carcinogenesis [164]. In the context of alcohol, the activation of the Il-6/STAT3 pathway in monocytes and other myeloid lineage cells, importantly promotes hepatic inflammation [165]. This pathway is tightly regulated by miRNAs [166] and conversely, IL-6 transactivates the expression of various miRNAs involved in liver diseases and HCC (e.g., miR-21) [84]. For instance, miR-223, which is upregulated in the serum and neutrophils of alcohol-fed mice, directly inhibits IL-6 and p47^phox^ expression, thereby attenuating ROS production and liver damage [70,71]. The STAT3 pathway is also regulated by miRNAs. MiR-29b, which is downregulated in macrophages (RAW264.7 and Kupffer cells from ethanol-fed mice), directly inhibits STAT3 [89].

### 4.5. Oxidative Stress

Oxidative stress importantly promotes hepatic inflammation during alcohol consumption [167]. MiR-214, which is upregulated in the liver of ethanol-treated rats and in a human hepatoma cell treated with ethanol, promotes oxidative stress by directly inhibiting the expression of glutathione reductase (GSR) and cytochrome P450 oxido-reductase (POR) [58]. Likewise, miR-34a is upregulated during ASH [79] and directly decreases the expression of SIRT1 [80,81], which plays a key role in protecting cells from oxidative stress [168]. In rats fed an LDC diet, miR-181b-5p expression is increased and directly targets PIAS1 (protein inhibitor of activated STAT1), a negative regulator of PRMT1 (protein arginine methyltransferase 1), which promotes oxidative stress and inflammatory response [75]. Finally, miR-219a-5p, which reduces ROS production by targeting the p66shc pathway, is downregulated in rats fed an LDC diet and in AML12 treated with ethanol [63].

### 4.6. Other Pathways

Other miRs, which are impacted by alcohol consumption, contribute to the development of steatohepatitis, through poorly characterized mechanisms. Some anti-inflammatory miRNAs are downregulated in the presence of alcohol, such as miR-199 [63], which directly reduces ethanol-induced expression of hypoxia-inducible factor 1-alpha (HiF-1α), thereby decreasing *monocyte chemoattractant protein-1* (MCP-1) release from Kupffer cells [64]. MiR-27a, which is upregulated by alcohol in monocytes from healthy subjects, promotes IL-10 secretion by directly targeting the ERK inhibitor Sprouty2 [76].

A decrease in miR-129-5p expression was observed in the serum of ASH patients and in alcohol-treated AML12 and ASH mice. This miR may suppress liver fibrosis by directly regulating the non-coding RNA long nuclear paraspeckle assembly transcript 1 (NEAT1) and suppressor of cytokine signaling 2 (SOCS2) [65]. This study underlines the importance of the interplay between miRNAs and lncRNAs in the development of ALD.

TGFβ-induced downregulation of miR-200a [57] has been associated with the severity of the disease by regulating the hedgehog pathway [60,169]. Indeed, miR-200a directly inhibits GLI family zinc finger 2 (Gli2), thus inhibiting the hedgehog pathway [59].

In the liver of mice fed an LDC diet, miR-27b, miR-214, miR-199a-3p, miR-182, miR-183, miR-200a, and miR-322 are downregulated, while miR-320, miR-486, miR-705, and miR-1224 are upregulated. However, the role of these miRNAs in ASH is still unknown [57], due to the lack of functional analyses. Finally, other miRNAs, detected in circulating extracellular vesicles of ASH mice, such as let-7f, miR-29a, and miR-340, have not been characterized yet, but may represent potent mediators of intercellular communication in the liver [87] and/or potential biomarkers of ASH.

### 4.7. ASH as a Priming Event of Hepatocarcinogenesis

ASH-related miRNAs are potentially paving the way for hepatic carcinogenesis by controlling key oncogenic processes. Some miRNAs, which are downregulated in ASH are well-known tumor suppressors, such as miR-122 [157], miR-200a [170], or miR-322, an inhibitor of galectin-3 [171]. In contrast, some miRNAs induced in ASH display potent oncogenic functions, such as miR-21, a well-established oncomiR [84], or other miRs involved in hepatic cancer cells proliferation (e.g., miR-155, miR-219a-5p, or let-7f) or invasion (e.g., miR-182) [172]. Together, these findings indicate that altered miRNAs in ASH may also prime the liver for carcinogenesis. Interestingly, this priming term is mostly used in the context of Non-Alcoholic Steatohepatitis (NASH), which is a major risk factor for HCC [173]. Targeting these priming alterations may represent an important chemopreventive approach to inhibit the progression of the disease toward HCC. However, such an approach might be limited by the early detection of ASH in patients, which is not associated with severe clinical signs.

## 5. MicroRNAs in Alcohol-Associated Cirrhosis

Continued alcohol consumption leads to the progression from steatohepatitis to alcoholic cirrhosis, which is characterized by hepatocyte damages and necrosis, replacement of liver parenchyma by fibrotic tissue, the appearance of regenerative nodules, portal hypertension, and a severe loss of hepatic functions [174]. Fibrogenesis is the main condition for the development of liver cirrhosis and thus activation of HSCs represent a key process in the development of cirrhosis [175]. HSCs are activated by cytokines released by several hepatic cell types (i.e., hepatocytes, Kupffer cells, endothelial cells) and are responsible for the activation of various signaling pathways (e.g., TGF-β1, PDGFα, LPS/TLR4, IL-6) [176]. TGF-β1 triggers HSCs trans-differentiation into myofibroblasts, which secrete important extracellular matrix components (e.g., COL1A1, αSMA, fibronectin) [177,178]. In parallel, IL-1β and TNF-α activate the NFκB pathway in HSCs, thereby ensuring their proliferation and survival [179]. The LPSs coming from the intestinal microbiota activates the TLR4 pathway [180], which in turn triggers HSC activation (e.g., upregulation of TGF-β1) but also the activation of Kupffer cells [150,181]. Activation of TLRs by LPSs activates the NADPH oxidase 1 (NOX1) complex, inducing the activation and proliferation of HSCs [182,183]. The role of miRNAs in the control of the different processes/pathways associated with hepatic fibrosis/cirrhosis is discussed below.

### 5.1. HSCs Activation

MiR-34a is upregulated in the liver of heavy drinker, as well as in the liver of LDC-fed mice [79]. MiR-34a promotes the proliferation, migration, and invasion of HSC and finally fibrosis by enhancing TGF-β1 [184] in ethanol-fed mice; it also inhibits HSC senescence, thereby fostering hepatic fibrosis [79]. Similar findings have been obtained in vitro on cultured hepatocytes treated with LPSs [79]. However, the direct mRNA targets of this miRNA were not clearly identified in this study. MiR-155 is another “fibromiR”, as evidenced by miR-155 KO mice, which are protected from alcohol-induced steatosis, inflammation, and fibrosis [48]. This effect is due to the ability of this miRNA to directly inhibit PPARγ, an anti-fibrotic protein, but also several other genes involved in fibrogenesis such as SMAD2/5, SNAIL1, or STAT3 [48]. In agreement, an increase in miR-155 expression has been documented in cirrhotic livers of alcoholic patients [48]. MiR-132, which is highly expressed in cirrhotic patients, is also an important promoter of hepatic fibrosis, as evidenced by an anti-miR-132 approach in a mouse model of fibrosis (CCL_4_-treated mice). Herein, the inhibition of miR-132 is associated with a decrease in pro-inflammatory and pro-fibrotic markers (e.g., COL1A1, αSMA, MCP1) and a decrease in caspase-3 activity in mice [74]. In contrast, miR-150 is downregulated in the serum and HSCs of rats and human patients with advanced ALD and act as an anti-fibrotic miRNA by reducing HSC activation (by inhibiting αSMA and *Col1A1* expression) [90].

In 2017, Satishchandran et al. showed an increase in grainyhead-like transcription factor 2 (GRHL2), an inhibitor of miR-122 expression, in cirrhotic patients and in the livers of alcohol-fed mice. Restoring miR-122 expression significantly reduces alcohol and CCL_4_-induced liver fibrosis [52]. The expression of miR-148a-3p is also decreased in rat models of alcoholic fibrosis. This miR directly targets the receptor tyrosine-protein kinase (ERBB3), and prevents apoptosis of HSCs by inhibiting BAX and the cleavage of caspase-3 [91]. Alcohol-induced downregulation of let-7 promotes Lin28 upregulation, which promotes HSCs activation [92]. Furthermore, alcohol exposure decreases miR-19b expression, an inhibitor of HSCs activation and proliferation [94]. MiR-19b directly targets TGFβRII and Methyl-CPG binding protein 2 (*MeCP2*), a critical epigenetic mediator of HSCs transdifferentiation [94]. Interestingly, the decrease in miR-19b expression is also coupled with an increase in pri-miR17-92 in HSCs. However, the role of pri-miR17-92 remains to be characterized [94] (Figure 4).

Similarly, other miRs deserve to be further characterized in the context of alcohol-related liver fibrosis, such as miR-181b [185], which promotes hepatic stellate cell proliferation, or miR-223 [69,70] and miR-214 [58], which are increased in ASH.

### 5.2. Hepatocyte Proliferation

Cirrhosis is defined by the appearance of regenerative nodules. This effect is mediated by the pro-inflammatory environment, and hepatocyte death and growth factors (HGFs), which trigger various signaling pathways responsible for hepatocyte proliferation (i.e., MAPK, c-fos, c-jun) [186]. During this step, hepatocytes coalesce into clusters, also known as nodules, which are surrounded by fibrotic tissue. These nodules can accumulate different mutations (e.g., p53, p21, c-myc, c-fos) and thus progress toward dysplastic nodules, thereby increasing the risk of hepatic carcinogenesis. This step requires an interplay between the different cell types of the liver. Among them, Kupffer cells secrete IL-6, which triggers the JAK/STAT3 signaling pathway in hepatocytes and promotes the transcription of cell cycle-related genes (e.g., c-fos, c-jun or c-myc) [187]. In addition, HSCs secrete hepatocyte growth factor (HGF), which initiates liver regeneration [188]. Finally, other pathways have been involved in hepatocyte cell proliferation and cirrhosis, including growth hormone (GH), insulin-like growth factors (IGF1 and IGF2), the PI3K/AKT pathway, somatostatin (SST), and MAPK signaling [186]. The impact of miRs on regenerative nodules in alcoholic cirrhosis remains poorly understood. Some miRNAs are known to importantly regulate these pathways but outside the scope of alcoholic cirrhosis, such as miR-29b, which suppresses the STAT3 pathway in ASH [89]; miR-100, which inhibits the IGF signaling in HCC [189,190]; and miR-101, which downregulates the PI3K/AKT pathway in HCC [191,192,193,194].

### 5.3. Other miRNAs with Poorly Characterized Functions

Several miRNAs are deregulated during hepatic regeneration and thus may contribute to the development of cirrhosis. For instance, miR-21 is induced during liver regeneration in a model of partial hepatectomy [85]. This effect is enhanced in alcohol-fed rats but the precise role of miR-21 in liver regeneration is still unclear [85].

Other studies have uncovered several miRs deregulated in the serum of patients with alcohol-related cirrhosis, including the induction of miR-486-5p, miR-92a-3p, miR-571, and miR-513-3p, and a decrease in miR-652 [95,96], as well as a decrease in miR-16 expression in exosomes [97]. Further studies are still required to characterize their roles and functions in alcoholic cirrhosis. Although the access to patient biopsies or sera represents an asset for the characterization of the disease, the lack of suitable in vivo models strongly limits our understanding of these miRNAs in cirrhosis. Indeed, the LDC diet with injections of LPSs [94] or CCL_4_ [52] allow for the development of steatosis, inflammation, and fibrosis, but does not allow the development of cirrhosis. In 2011, Yip-Schneider et al. developed a model of cirrhosis in rats fed with alcohol for 18 months. These animals showed liver damage and the appearance of regenerative nodules [195].

### 5.4. MiRNAs Fostering HCC Development

Hepatic cirrhosis represents an important risk factor for hepatocarcinogenesis, due to the accumulation of mutations in hepatocytes [196]. However, this transition is not only a matter of genetic damage since several miRNAs are deregulated at this step and play a role in cancer-related processes. The miR-17-92 cluster, which is upregulated in cirrhosis, is a well-characterized oncomiR due to its capacity to inhibit the expression of cAMP Responsive Element Binding Protein Like 2 (*CREBL2*), Proline Rich and Gla Domain 1 (*PRRG1*), and Netrin 4 (*NTN4*) [197,198]. MiR-132 is also overexpressed in cirrhosis and in HCC, and correlates with a higher tumor grade and stage and a poor clinical outcome [74]. Alteration of the let-7/Lin28 axis has also been demonstrated during the development of HCC [92]. Let-7 is a tumor suppressor, which inhibits the Wnt/β-catenin signaling pathway, thus preventing the self-renewal of HCC stem cells [199]. Another example is miR-148a-3p, which is downregulated in cirrhosis, and inhibits ERBB3, a proto-oncogene [200]. Others have a protective role, such as miR-486-5p, whose expression is increased in patient sera and exerts tumor suppressive functions [201]. In 2020, Felgendreff et al. highlighted 50 miRs whose expression changed between tumor-free cirrhosis and hepatocellular-associated cirrhosis in alcoholic patients [202]. Around forty miRs were identified in the livers of cirrhotic patients as compared to healthy patients (Figure 5A); among them, some have previously been associated with tumor-promoting functions, while others inhibit HCC development (Figure 5B). In this context, it is likely that the progression toward HCC is determined by an imbalance between pro- and anti-tumorigenic alterations. Deciphering the mechanisms responsible for this disequilibrium may offer novel therapeutic perspectives. Of note, most deregulated miRNAs of this study have not been associated with HCC yet (Figure 5C), and thus may represent new oncomiRs or miR-suppressors, such as miR-3622a, which is strongly induced in HCC (Figure 5D). Finally, a comparative analysis of miRNAs deregulated in alcohol-induced cirrhosis and NASH-induced cirrhosis revealed very few similarities (Figure 5E). These findings suggest a distinct miRNA-specific signature promoting HCC development in these two different contexts.

## 6. MicroRNAs in Alcoholic Hepatitis (AH)

Alcoholic hepatitis (AH) represents an acute and severe hepatic inflammation [50] characterized by a wide range of pathological features, including hepatocyte degeneration and ballooning, a ductular reaction, cholestasis, neutrophil infiltration, the secretion of pro-inflammatory cytokines (e.g., TNF-α, IL-1β, IL-6, and IL-8), alteration of the gut permeability (translocation of LPS to the liver [203]), and the accumulation of protein aggregates called Mallory–Denk bodies in hepatocytes [129,204]. In 90% of cases, AH occurs in the context of hepatic cirrhosis but can also occur from earlier stages such as ASH [205]. Patients suffering from AH display several clinical symptoms including jaundice, hepatic encephalitis, and bleeding from the gastrointestinal tract [36]. Unfortunately, AH is associated with a high mortality rate (40% within 6 months of onset of clinical syndromes) [206], due to a severe hepatic insufficiency, a limited number of therapeutic options, and the resistance to corticoids [20]. To date, only liver transplantation can provide a cure to patients [207]. Deciphering the molecular bases of AH is therefore of major interest to develop new and efficient therapeutic options and/or to alleviate the resistance to current treatments (corticoids).

### 6.1. Hippo/Yes-Associated Protein (YAP) Pathway Ductular Reaction

AH is characterized by an impaired liver regeneration, which is tightly associated with an inhibition of the Hippo signaling in hepatocytes [208]. In AH patients, this effect has been attributed to a decrease in Macrophage stimulating 1 (MST1) expression, which triggers the trans-differentiation of hepatocytes into cholangiocytes, thereby increasing the ductular reaction [208]. Although the role of miRNAs in the regulation of the Hippo/YAP pathway has been highlighted in the context of HCC (e.g., miR-15b, miR-130, miR-21-3p) [209], this link has not been investigated yet in AH. Interestingly, the ductular reaction further enhances hepatic inflammation by increasing the expression of miR-182 in biliary cells [40]. Interestingly, the overexpression of miR-182 correlates with the ductular reaction, the disease severity, and a high mortality [40].

### 6.2. TLR and NFκB Signaling

Overexpression of Let-7 was also observed in alcohol-fed mice and in patients with AH. Let-7 is also secreted (e.g., let-7b) and binds to TLR-7, thus activating the MyD88/NFκB pathway and triggering an important inflammatory response [93]. MiR-182 is also increased in AH patients and mouse models (e.g., ethanol intake, CCL_4_, and ethanol + CCL_4_ model), and promotes inflammation (*Mcp-1, Ccl20, Cxcl5, Cxcl1*) and anti-apoptotic (*Bcl2*) genes [40]. Alcohol exposure leads also to a decrease in miR-148a by decreasing Forkhead box protein O1 (*FoxO1*). MiR-148a directly targets and inhibits thioredoxin-interacting protein (*TXNIP*), a protein activating the NOD-like receptor family, pyrin domain containing 3 (NLRP3) inflammasome, and caspase-1-induced pyropoptosis [100]. During AH, miR-30e expression is also downregulated and this effect correlates with an increase in Uncoupling protein-2 (UCP2), but also inflammation, and a decrease in ATP and H_2_O_2_ levels [101]. MiR-21, which is upregulated in HSCs during AH, importantly promotes the NFκB pathway by directly targeting the 3′ UTR of Von Hippel–Lindau (VHL) [86]. In 2015, Yin et al. demonstrated that miR-217 is increased in alcoholic hepatitis [45], in mouse livers, macrophages, and Kupffer cells exposed to ethanol and LPSs. MiR-217 directly inhibits SIRT-1, an inhibitor of NFκB and nuclear factor of activated T-cells 4 (NFATc4) activity [45].

### 6.3. Circulating microRNAs

Secreted miRNAs importantly contribute to intercellular crosstalk [210,211] and the regulation of several physiological and pathological processes [212,213], including inflammation [153]. Moreover, circulating miRNAs can be detected in body fluids and thus may represent novel non-invasive biomarkers for a wide range of human diseases [214]. In the presence of alcohol, primary human monocytes secrete extracellular vesicles (EVs), which promote anti-inflammatory macrophages M2-polarization. This effect is mediated by miR-27a, which is contained within these EVs and targets CD206 [77]. Finally, an increased number of EVs with a high level of miR-27a and miR-181 was also detected in the plasma of patients with AH [77]. Both miRs were found to be upregulated in EVs derived from mouse hepatocytes mimicking alcoholic hepatitis. When transfected into HSCs, mir-27a and miR-181 repressed nuclear receptor subfamily 1 group D member 2 (Nr1d2), a marker of quiescent HSCs [78].

Other secreted miRs, such as let-7 by hepatocytes, trigger a major inflammatory response by binding to TLR-7 and activating the MyD88/NFκB pathway when alcohol is consumed [93]. Interestingly, several other miRs, inhibiting hepatic inflammation and fibrosis are upregulated in the sera and exosomes of AH patients [53], including miR-122 and miR-30a [102]. MiR-291b, which is upregulated in the sera and exosomes of AH patients, inhibits the expression of Toll-interacting protein (Tollip), a negative regulator of the MyD88-dependent signaling in rat Kupffer cells [103]. Further studies are required to determine whether these alterations are causative of AH or simply a consecutive defense response against severe inflammation. Indeed, potentiating the effect of protective miRNAs may represent an efficient strategy to resolve severe inflammation. Moreover, these circulating miRNAs may also represent efficient biomarkers from liquid biopsies, unless they are unspecific to AH, as compared to other hepatic/inflammatory diseases.

Finally, several other circulating miRNAs have been found increased in the plasma of AH patients and correlate with poor prognosis, such as miR-30b-5p, miR-20a-5p, miR-146a-5p, and miR-26b-5p [99] or miR-155 [50]. Similarly, the analysis of EVs from the serum of alcohol-fed mice and AH patients, revealed an increase in miR-122, miR-192, and miR-30a [53]. However, these studies remain strongly descriptive and intense efforts are still required to understand the functions of these miRNAs.

### 6.4. Other miRNAs

The expression of miRs will also affect other mechanisms during AH. An increase in miR-34a and a downregulation of miR-483-3p could explain the various mechanisms of Mallory–Denk body formation and inhibition of cell regeneration. Because miR-483-3p inhibits breast cancer 1 (BRCA1) expression, its overexpression may impair cell cycle progression [82]. Other miRs also act on cell death, such as miR-150-5p, which is overexpressed in the livers of AH patients and inhibits the E3 ligase cytokine-inductible SH2-containing protein (CISH), thereby increasing the expression of Fas-associated protein with death domain (FADD). The increase of FADD activates caspase-3 and enhances apoptosis [104].

In another study, an increase of 111 miRNAs, including miR-182, miR-21, and miR-214, and a decrease of 66 miRNAs (including miR-422a) has been observed in the liver of AH patients [40]. Among them, miR-182 expression correlates with the ductular reaction and a poor clinical outcome in patients [40]. Overexpression of miR-182 (using a mimic oligonucleotide) in cholangiocytes promotes the upregulation of pro-inflammatory and cell cycle-related genes (*CCL20*, *CXCL1*, *Il-8*, and *Cyclin D1*). However, this study remains descriptive and intense efforts are still required to understand the functions of these other miRNAs.

Taken together, these findings indicate that miRNAs are strongly involved in AH. However, due to the lack of in vivo models recapitulating the alterations observed in patients our knowledge of miRNA function in AH is strictly limited to in vitro models (cell lines and primary cells). Developing new models of AH represent one of the most important challenges in the field.

## 7. MicroRNAs in Alcohol-Related Hepatocellular Carcinoma (HCC)

Intense efforts have been devoted to characterize HCC at the genetic levels [215]. However, it is now clear that epigenetic defects importantly contribute to the altered expression of oncogenes, drivers or tumor suppressors, or tumor-promoting processes (e.g., chronic inflammation) [216]. The role of miRNAs in hepatocarcinogenesis has been well documented and miRNAs, importantly, control the most common cancerous hallmarks but also the pathways associated with hepatocarcinogenesis [217,218]. In agreement, suppression of miRNA processing machinery genes like Dicer, DGCR8, Drosha, and transactivation response RNA binding protein (TRBP), reduces miRNA maturation and synthesis and leads to HCC development [219,220]. However, most studies characterizing miRNAs in HCC are using models unrelated to alcohol etiology. Mouse models are commonly used to study HCC but their aversion and higher alcohol metabolism compared to humans make ethanol-enriched diet models insufficient to develop HCC without genetic engineering, implantation, or chemical induction [221]. Other models of cirrhotic HCC exist like transgenic oncopig cancer models undergoing ethanol infusion to develop concomitant fibrosis [222]. Finally, as discussed before, alcohol-associated cirrhosis involves strikingly different miRNAs as compared to NASH-associated cirrhosis, thus indicating that the mechanisms fostering HCC is also different. In this chapter, we are therefore focusing on miRNAs in the context of ALD-associated HCC.

### 7.1. miRNAs with Oncogenic/Tumor Suppressive Functions in ALD-Related HCC

Although the importance of miRNAs in HCC development is well-established [217], our knowledge is limited to models unrelated to chronic alcohol consumption. Whether these miRNAs are also involved in alcohol-related HCC is not guaranteed. Deciphering the specific miRNA signature in alcohol-related HCC is therefore of major importance to identify new biomarkers and/or therapeutic targets.

A bioinformatic analysis by Shen et al. on 48 human HCC tumors, identified the upregulation of four miRNAs, including miR-10b, miR-21, miR-500a, and miR-532 [223] and the downregulation of eight miRNAs including miR-424, miR-3607, miR-24-1, miR-139, miR130a, miR-29c, miR-101-1, and miR-101-2 in the context of alcohol abuse [223]. Although these miRNAs were previously associated with HCC-related processes [191,224,225,226,227,228,229,230,231], their role in alcohol-related HCC remains unexplored. MiR-21 is a well-established oncomiR in HCC [232,233] and its expression is also increased in alcohol-treated hepatic cancer cells (HepG2) [84]. Upon ethanol treatment, IL-6 induced STAT3 activation, which binds to miR-21′s promoter and increases its expression. In turn, miR-21 promotes cancer cell survival. However, the induction of miR-21 in patients with alcohol-associated HCC does not correlate with patient prognosis, [234], thus contrasting with other studies in “non-alcoholic HCC” [235,236]. Surprisingly, recent findings have demonstrated that the loss of miR-21 in hepatocytes in vivo promotes hepatic carcinogenesis in a model of diethylnitrosamine-treated mice [237], thus suggesting that miR-21 can also exert tumor suppressive properties. The literature is therefore providing discrepant information regarding miR-21′s functions and thus further studies are required to evaluate the therapeutic potential of targeting miR-21 in suitable in vivo models of alcohol-related HCC. A miRNA profiling of human HCC tumors revealed that miR-126* is downregulated in alcoholic HCC [238]. The consequences of this downregulation remain to be investigated in the context of alcohol-induced HCC.

Several factors, like DNA methylation, hypoxia, or endogenous factors (stress, steroid hormones) are known to regulate the expression of miRNAs [239]. Among them, β-catenin, one of the main alterations in HCC [240], is activated by ethanol exposure in HepG2 cells [108] and induces miR-22-3p expression. In turn, miR-22-3p promotes HCC by directly downregulating Ten-eleven-translocation 2 (TET2) expression [108].

Finally, in a mouse model of alcoholic HCC (Lieber–DeCarli alcohol diet + intraperitoneal injection of DEN), miR-122 expression is downregulated [54], thus leading to the overexpression of cyclin G1 and hypoxia-inducible factor 1-alpha (HIF1α) expression, two direct targets of this miRNA involved in cancer cell proliferation and invasion [54].

### 7.2. Other miRNAs with Poorly Defined Functions in ALD-Related HCC

The role of miRNAs in alcohol-related HCC is largely underestimated. Based on the literature (Table 2), these miRNAs are involved in the regulation of oncogenes (e.g., miR-15a and wnt3a), tumor suppressors (e.g., miR-191 and KLF6), as well as several pathways associated with hepatocarcinogenesis (see example in Figure 6B). In a transcriptomic dataset (GSE10694), we cross-compared alcohol-related HCC with non-tumoral livers (Figure 6A). This analysis revealed a whole set of differentially expressed miRNAs between the two groups, including 10 being upregulated (miR-106a, miR-106b, miR-15b, miR-191, miR-210, miR-221, miR-222, miR-224, miR-25, miR-331) and 11 downregulated (miR-100, miR-101, miR-10a, miR-125b, miR-148a, miR-15a, miR-199a, miR-199a*, miR-22, miR-422b, miR-99a) in the tumors as compared to healthy controls. Based on the literature (Table 2 and Figure 6B), these miRNAs are involved in several HCC-related pathways. Of note, some miRNAs exert pleiotropic functions on several pathways and thus represent potential therapeutic targets. Our analysis is limited by the sample size but it still gives an indication about a possible miRNA profile of HCC with alcohol abuse. Such profiles can be used as a starting hypothesis for future studies to be validated at first and may be used as a diagnostic tool.

Other bioinformatic studies also revealed that miR-432, whose expression is increased in the ASH mouse model (LDC diet) could be a predictive biomarker for HCC [241]. Besides these miRNAs, several have been identified in HCC from alcohol abusers infected with HBV, such as miR-223 and miR-944, which are upregulated in alcohol-associated HCC. Other miRNAs, such as miR-9 and miR-153-2-p, are downregulated in the HBV-positive HCC drinkers group compared to the HCC non-drinkers group [242].

In a study gathering 186 North American patients, miR-26a is downregulated in HCC tumors from patients with chronic alcohol consumption compared to adjacent non-tumor tissues [110]. However, the role of miR-26 in alcoholic HCC remains to be investigated.

Taken together, these data indicate that the miRNAs deregulated in ALD-related HCC have been largely underestimated. Very few in vivo models are available to study ALD-related hepatic carcinogenesis. Although ethanol exposure (Lieber–DeCarli Diet) in mice can accelerate hepatic carcinogenesis induced by diethylnitrosamine [243], this model does not fully recapitulate the features of ALD-related HCC in patients. New models are urgently needed to perform functional analyses of miRNAs in this disease. Moreover, other aspects underlying the complexity of miRNA-dependent regulation should be considered. The presence of a single nucleotide polymorphism (SNP) in an miRNA sequence may alter miRNA expression and influence hundreds of target genes, as suggested for a SNPin the promoter region of pri-miR-34b/c, which correlates with an increased risk of developing HCC in patients with a history of alcohol abuse [244].

**Table 2 cancers-15-05557-t002:** Summary of deregulated miRNAs and their impacts on different pathways associated with HCC.

MiRs	Pathways	Model	Function	Target	Ref
**MiR-100**	PI3K/AKT/mTOR IGF signaling	HCC cells from patients, Human HCC cell lines (SK-Hep1, MHCC97-L, SMMC-7721, HCCLM3, Huh7, Hep3B, and HepG2),	Tumor growth inhibition Apoptosis promotion Autophagy induction	Insulin-like growth factor 2 (IGF2), mammalian target of rapamycin (mTOR), and insulin like growth factor 1 receptor (IGF-1R)	[189,190]
**MiR-101**	PI3K/AKT/mTOR, TGFβ, Epigenetics	HBV-related HCC tissue from patients, immortalized liver cell line L-02, and human HCC cell lines (HepG2, Hep3B, SMMC-7721, Huh7, MHCC-LM9)	Autophagy inhibition, Invasion and EMT inhibition, proapoptotic function, prevention of HCC progression	mTOR, EZH2, H3K27me3, EED, myeloid leukemia cell differentiation protein (Mcl-1), DNA methyltransferase 3A (DNMT3A), TGFβR1, Smad2	[209,228,230,245,246,247,248,249]
**MiR-106a**	TP53/Cell cycle	Human HCC cell lines (HepG2 and Hep3B)	Apoptosis resistance, cell cycle progression and invasion	Tumor Protein P53 Inducible Nuclear Protein 1 (TP53INP1) and cyclin dependent kinase inhibitor 1A (CDKN1A)	[250]
**MiR-106b**	TP53/Cell cycle, TGFβ signaling	Tissue from patients, Human HCC cell lines (Hep3B, Huh7, HepG2, and Bel-7402)	Promote HCC cell proliferation and migration	Disabled homolog 2 (DAB2), SMAD Family Member 7 (SMAD7)	[207,209]
**MiR-10a**	PI3K/AKT/mTOR	HCC patients, human HCC cell lines (Huh7, HepG2, and PLC)	Cell proliferation inhibition chemosensibility	Musashi 1 (MSI1)	[251]
**MiR-125b**	IL6/JAK/STAT, IGF signaling, Apoptosis, epigenetics	Human HCC cell lines (MHCC97L, SMMC7721, HepG2, HL-7702), HCC tissue from patients	Promote apoptosis, induce cell senescence and invasion inhibition	IGF2, Mcl-1, Bcl-w, Interleukin (IL)-6, IL-6R, sirtuin 6 (SIRT6) and SIRT7	[189,252,253,254]
**MiR-148a**	Epigenetics, PI3K/AKT, TGFβ signaling	Human HCC cell lines (MHCC97, Huh7, HepG2, SMMC-7721, and HCCLM3), normal liver cell line L02	Cell proliferation inhibition Cell migration and invasion inhibition	DNA methyltransferase DNMT1, Death receptor-5 (DR-5), SMAD2	[240,241,255]
**MiR-15a**	WNT/β-catenin, TGF-β signaling, epigenetics, JAK/STAT	HCC tissue from patients, Human HCC cell lines (HCC-LM3, Huh-7, CSQT2, HepG2, MHCC97H, and SMMC-7721), normal liver cell line THLE2, Tumor xenograft	Inhibition of HCC proliferation, migration and invasion. Promote apoptosis	O-linked N-acetylglucosamine (GlcNAc) transferase (OGT), Transforming Growth Factor Beta 1 (TGF-β1), SMAD7, WNT3A, signal transducer and activator of transcription 3 (STAT3)	[256,257,258,259,260]
**MiR-15b**	Apoptosis, WNT/β-catenin	HCC patients, Human HCC cell lines (HepG2, Huh7, Hep3B, MHCC-97L and MHCC-97H)	Cell proliferation inhibition Promote apoptosis	WNT3A, B-cell lymphoma 2 (BCL-2)	[261]
**MiR-191**	TP53/Cell cycle	HCC tissue from patients, Hep3B and HepG2 cell lines	Cell cycle progression and cell proliferation	ZO-1-associated Y-box factor (ZONAB)/cyclinD1	[262]
**MiR-199a**	HGF/c-Met, TP53/Cell cycle, PI3K/AKT/mTOR	Human HCC cell lines (Huh7, HepG2, SNU182, PLC/PRF/5, Hep3B, SNU423, and SNU449)	Inhibition of cell proliferation, cell cycle arrest, apoptosis induction	CD44, mTOR, c-Met, zinc-fingers and homeoboxes-1 (ZHX1)	[263,264,265]
**MiR-210**	PI3K/AKT	Human HCC cell lines (HepG2, MHCC-97H and HuH7)	Promote proliferation and invasion Inhibition of apoptosis	PI3K, AKT, mTOR	[223]
**MiR-22**	*Epigenetics*, TP53/Cell Cycle	HCC tissue from patients, Human HCC cell line PLC/PRF/5 and MHCC97L	Induction of apoptosis Cell proliferation inhibition	X-linked IAP (XIAP), Histone deacetylase 4 (HDAC4), Cyclin-dependent kinase inhibitor 1A (CDKN1A)	[266,267,268]
**MiR-221**	PI3K/AKT/mTOR TP53/Cell cycle	HCC patients, HCC cell lines (PLC/PRF/5, Huh7, HepG2, SNU-449, SNU398, SNU-423 and SK-Hep-1)	Cell proliferation Cell cycle progression	CD44, CDKN1B/p27, CDKN1C/p57 DNA damage-inducible transcript 4 (DDIT4)	[228,229,230]
**MiR-222**	PI3K/AKT/mTOR, TP53/Cell Cycle	Human HCC cell lines (HepG2, Hep3B, HKCI-4, and HKCI-9)	Cell proliferation, Migration, and invasion and inhibits apoptosis	p27 protein phosphatase 2A subunit B (PPP2R2A)	[269,270]
**MiR-224**	PI3K/AKT/mTOR, TGFβ signaling	HCC tissue from patients, Human HCC cell lines (HepG2)	Cell proliferation	Protein Phosphatase 2 Scaffold Subunit Abeta (PPP2R1B), SMAD4	[271,272,273]
**MiR-25**	WNT/β-catenin	Human HCC cell lines (HCCLM3 and Huh7)	cell proliferation, migration and invasion	PTEN	[274]
**MiR-99a**	IGF signaling, TP53/cell cycle Epigenetics	HCC tissue from patients, Human HCC cell lines (Hep2G SMMC-7721, Huh7, and Hep3B)	Cell proliferation and invasion inhibition, block cell cycle	IGF1R mTOR AGO2	[275,276,277]

## 8. Therapeutical Strategies against ALD/HCC-Related miRNAs

### 8.1. A Myriad of Strategies to Target miRNAs

Regulating miRNAs to shape the transcriptome is a promising therapy for ALD. Based on the miRNA landscape of ALD and HCC, several miRNAs may represent therapeutic targets. Inhibiting the detrimental miRNAs, or instead restoring the protective ones, could be achieved using different strategies (Figure 7).

Downregulated expression of beneficial miRNA can be restored by the intracellular delivery of miRNA mimics, agomiRs, or plasmids encoding miRNAs. In contrast, strategies have been designed to decrease the expression of overexpressed detrimental miRNAs. These miRNA suppression therapies are based on the nucleotide complementarity between the miRNAs and anti-miR oligonucleotides (AMO), like miRNA inhibitors, antagomirs, miRNA masks, small RNA zippers, ceRNA (competing endogenous RNA), and miRNA sponges. This latter being designed to bind and compete for the binding of several miRNAs to their mRNA targets, which is a similar strategy to that developed with circular-RNA [278,279]. Other opportunities rely on gene-editing systems like CRISPR/Cas or using small-molecule inhibitors or degraders (SMIR) [280,281]. However, these therapeutical opportunities face several issues, especially when administered intravenously: poor pharmacodynamics (degradation by RNAse, rapid blood clearance), non-specificity of the miRNA delivery to the biological target, low tissue permeability, and physical properties making the miRNAs unable to enter cells in their native form. In this context, many chemical modifications have been performed on nucleotides or the phosphoribosyl backbone to improve miRNA efficacy and half-life [278].

To avoid miRNA degradation from the administration site and to improve the tissue specificity, increasing efficiency while decreasing the side-effects of miRNA-based therapeutics, carrying vehicles have been developed [282], such as lentivirus (LV), retrovirus (RV), adenovirus (Ad) and Ad-associated viruses (AAV) [278], and virus-like particles (VLP) [283,284,285,286]. While RV and LV can express miRNA mimics or antagomir over long periods of time due to their genomic integration, this random process could be critical for the cells. Ad and AAV are interesting but immune reactions have been reported both in rodent models and humans [287,288], and further efforts must be made to unlock their full potential as miRNA delivery systems. Non-viral-based delivery systems involving nanocarriers (NCs) and modified extracellular vesicles (EVs) may represent an alternative option. Firstly, EVs, or exosomes, are 50–300 nm vesicles secreted by cells containing biological compounds including miRNAs. These natural carriers are produced and enriched for miRNA ex-vivo using mesenchymal or adipose-derived stem cells as biofactories. While limitations prevail, EVs constitute the most promising opportunity for the safe targeted delivery of miRNA regulators [289]. Finally, among their extreme diversity, lipid-based and polymeric delivery systems represent the most used NCs with a size range below 250 nm [282,290,291,292].

In this general context, delivering miRNA regulators to the liver appears possible. Clinical successes from the hepatic delivery of siRNA encourage the miRNA therapy [293]. In the following paragraph, we discuss the potential miRNAs that could be targeted for the treatment of ALD and HCC.

### 8.2. Therapeutic targeting of miRNAs in ALD

#### 8.2.1. Steatosis

Targeting miRNAs to prevent alcohol-induced steatosis may represent an interesting approach to avoid progression toward more severe stages of the disease (i.e., fibrosis). However, it should be kept in mind that hepatic steatosis is a protective mechanism against detrimental free fatty acids (e.g., palmitate) [294,295]. Thus, impairing miRNAs involved in de novo lipogenesis may reduce hepatic steatosis but may worsen lipotoxicity and thus hepatic fibrosis. This effect has been documented for several strategies aiming at impairing de novo lipogenesis [296].

#### 8.2.2. ASH

Targeting deregulated miRNA in ASH is also interesting, given that these miRNAs are not only pro-inflammatory but are also priming the liver for hepatocarcinogenesis. Moreover, some miRNAs display pleiotropic regulatory functions on the pro-inflammatory processes of ASH (e.g., miR-155). Targeting HCC priming events may reduce the occurrence of hepatic tumors in alcoholic patients. Based on our literature overview, few miRNAs could be targeted, including miR-122 or miR-21. An elegant strategy aimed at sponging miR-21 while delivering pre-miR-122 in HCC has recently been developed in vitro [285] and may pave the way for in vivo assay in ASH models. Other have described the hepatic delivery of miR-122 for the treatment of HCC in mouse using lipid nanocarriers or exosomes that can be repurposed in ASH to limit the occurrence of HCC [297,298]. However, such approaches were never investigated in the context of ALD. Moreover, it also remains to develop more physiological models of ASH. To date, the Lieber–DeCarli diet + CCl_4_ is the only model allowing hepatic steatosis and inflammation.

#### 8.2.3. Cirrhosis

In alcoholic cirrhosis, we have discussed several miRNAs that could be targeted by specific strategies. However, few of them have been evaluated as potential therapeutic targets. Among them, the inhibition of miR-132 by intraperitoneal injection of LNA-anti-miR-132 efficiently reduces hepatic fibrosis in CCl_4_-treated mice [74]. Although these preclinical findings are encouraging, further efforts are still required to characterize the therapeutic potential of targeting these miRNAs.

#### 8.2.4. HCC

Given the wide range of miRNAs involved in alcohol-related HCC, it might be difficult to make a choice and target only one miRNA. Targeting multiple miRNAs may represent an appealing approach, but another strategy could be to target miRNAs with the most pleiotropic functions on HCC-related pathways. In that sense, miR-191 and miR-222 may represent potential targets due to their capacity to control several pathways, including TP53, and the Wnt/β-catenin and PI3K/AKT signaling.

To specifically address the miRNA described in this review, one should keep in mind the complex cellular interplay in the liver. Indeed, ALD cannot restrict to the sole parenchymal hepatocytes but instead the surrounding non-parenchymal cell types must be considered like the hepatic stellate cells (HSCs) [299] and the Kupffer cells (KCs) [300,301]. A safe and efficient miRNA-based delivery system should possess a passive targeting property or have a targeting moiety toward one of those liver cells types (designated as an active targeting), to avoid adverse side effects as described in the MRX34 (miR-34a mimics) phase I clinical trial [302]. Negatively charged NCs can be opsonized in the blood flow and, together with a size larger than 100–200 nm, they are easily taken up by the liver sinusoidal endothelial cells (LSECs) and the KCs. Hydrophobic NCs are likewise more quickly captured by these cells [303]. On the other hand, smaller NCs can reach the space of Disse through the LSEC fenestrations and thus the HSC and the hepatocytes, especially if they have been decorated with poly-ethylene glycol (PEG) to improve their stealthiness and escape the immune surveillance [303,304]. However, this passive targeting is not sufficiently precise to target one liver cell type and targeting a moiety is recommended for that purpose, as previously reviewed [291,292] and summarized here in Figure 7.

### 8.3. Therapeutic Approaches Targeting miRNAs in Clinical Trials and Future Perspectives

To date, no miRNA suppression or replacement strategy using an active targeted delivery system exists in the therapeutic arsenal despite the promise of success. MiRNA-based clinical trials, investigational miRNA-based therapies, patented, approved, or marketed medicine have been reviewed [279,282,305,306]. Although there are currently no clinical trials on miRNAs targeting in ALD, some miRNAs involved in ALD or ALD-related HCC have been studied in other contexts (see examples in Table 3). Few clinical trials have been devoted to HCC or Hepatitis C virus (HCV), such as miravirsen (anti-miR-122), MRX34, or RG-101 (anti-miR-122), which have not yet passed clinical trial phase I/II [307,308,309]. MRX34 has even been halted because of off-target delivery of the miRNA mimic [310]. However, very sparse data are published on miRNA as a clinical target to treat the consecutive ALD stages before the occurrence of HCC.

Some carriers have been developed, encouraging further research. For example, a miR-122 lipoplex consisting of a cationic lipid nanoparticle formulation allowed miR-122 hepatic delivery and restored deregulated gene expression in the HCC mouse model [311]. Other lipid-based or polymeric nanocarriers for hepatic miR-122 delivery demonstrated a hepatic tropism but without actively targeting a specific cell type [297,312]. Adding a targeting moiety ameliorates the efficacy of the miRNA delivery, like the use of GE11 (targeting the EGF Receptor overexpressed in HCC) decorated Virus-like Particle for sponging miR-21 together with the delivery of a miR-122 mimic [285]. Other have described a galactosylated-chitosan NC to deliver miR-122 that sensitized HCC cells to a co-delivered anticancer drug [313]. In a context of steatosis and HCC, exosomes genetically modified to express anti-miR-199a-5p or miR-223 [314,315], lactosylated-polymeric methacrylate-based NC loaded with miR-146b mimic [316], or anti-glypican3-decorated liposome loaded with an anticancer and anti-miR-27a [317] are other examples of hepatocyte-targeted miRNA delivery systems. Besides those, the use of small molecules could be of interest as shown in a mouse model of ALD in which Baicalin-stimulated expression of miR-205 led to the inhibition of NF-kB-driven inflammation and finally protected the liver against ethanol-induced injury [88]. This latter strategy could be enhanced by the use of hepatic-targeted carriers and assessed for its ability to limit HCC occurrence. Focusing on HSC reveals that the main targeting strategy exploits the affinity of these cells for the retinol binding protein with liposome loaded with vitamin A and miRNA [318,319], and a clinical trial to deliver oligonucleotides to HSC using Vitamin A (NCT02227459). Finally, passive targeting is used for miRNA delivery in KCs, as described by Liu et al. in mice [320], where a polymeric carrier with a diameter of 279 nm and a positive charge serves as a synthetic anti-NFkB miRNA delivery platform. However, one can fear off-target side effects as for MRX34 [302]. More recently, NCs have been developed to target both KCs and HSCs and disrupt their detrimental crosstalk in ALD, especially to reverse liver fibrosis. The two reported strategies relied on polymeric NCs able to deliver anti-miR-155 to KCs in parallel with the blockade of the HSC’s CXCR4. Cyclam derivatives, known to inhibit CXCR4 [321], decorated polyethylene imine core NCs loaded with anti-miR-155. With sizes of 60 and 150 nm, respectively, and a positive surface charge, both NCs reversed the hepatic damages in an ethanol/CCl4 mouse model of liver fibrosis [322,323]. Finally, starting from an amino–lipid-based nanocarrier library, it has been demonstrated that the surface of the nanocarriers can be functionalized by blood circulating proteins to obtain an active targeting of the liver cells. Depending on the amino–lipid, NCs were decorated by a corona of either apolipoprotein E or albumin, leading to the targeting of the hepatocytes or KCs, respectively [324], while sharing similar physical properties. These carriers have proven efficient in the delivery of let-7 g miRNA in an aggressive myc-driven HCC mouse model.

## 9. Conclusions

Although ALD is the most prevalent liver disease in developed countries, there are currently no reviews documenting the role of miRNAs in the all the stages of this disease. Our study is not only providing an exhaustive overview of the role of miRNAs in the development of ALD but also provides evidence that deregulated miRNAs at each stage of the disease contribute to the establishment of a neoplastic phenotype. More than one hundred miRNAs are discussed, thus highlighting the importance of post-transcriptional regulation of gene expression in ALD and HCC and raising many questions regarding the therapeutic targeting of these miRNAs. Currently, they are no miRNA-targeted delivery systems for the treatment of ALD on the market. Although many strategies can be designed to efficiently target these miRNAs, it remains to be determined which ones should be targeted. Moreover, more suitable in vivo models are tremendously required to characterize the role of these miRNAs in ALD/HCC and evaluate the potential of their therapeutic potential. The very first stages of ALD, including steatosis, are not a primary source of research and the targeted delivery of miRNA mainly focuses on the later stages like fibrosis resolution or HCC remission. The development of dual therapeutics, combining several drugs (anti-miR and anticancer) or targeting several cell types (KCs and HSCs), together with a passive-to-active targeting, pave the way for efficient future treatments of ALD. Furthermore, increasing evidence challenges the dogmatic view of miRNAs as strict inhibitors of gene expression, and suggest, in contrast, that miRNAs can induce gene expression [325]. Finally, it should be remembered that miRNA-dependent regulation is a complex process tightly regulated by other trans-acting factors (e.g., lncRNAs or RBPs), which regulate the bioavailability and the activity of miRNAs. Emerging evidence indicates that this interplay is relevant in ALD, as shown by miR-214, which is sponged and inactivated by the ethanol-induced lncRNA urothelial cancer-associated 1 (*UCA1*) in a hepatocyte cell line [326]. The complexity of miRNA-dependent functions is further enhanced by miRNAs editing by specific enzymes (e.g., Adenosine Deaminase, RNA specific, ADAR) controlling miRNA functions and whose expression is often imbalanced in pathological states (i.e., HCC) [327,328].

## Figures and Tables

**Figure 2 cancers-15-05557-f002:**
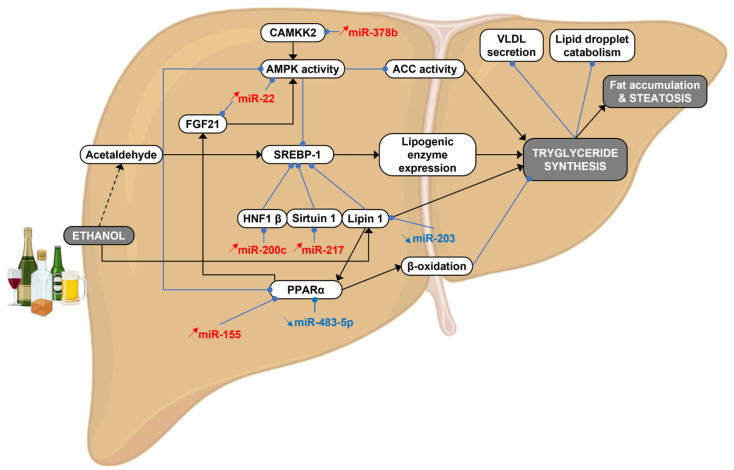
Impact of different microRNAs on ethanol metabolism inducing fat accumulation and steatosis within the liver. Black arrows: activation of the pathways. Blue arrows: pathway inhibition. FGF21: fibroblast growth factor 21; CAMKK2: Ca^2+^/calmodulin-dependent protein kinase kinase 2; AMPK: AMP-activated protein kinase; SREBP-1: sterol regulatory element binding protein 1; HNF1β: hepatocyte nuclear factor 1 homeobox β; PPARα: peroxisome proliferator-activated receptor α; ACC: acetyl-coA carboxylase; VLDL: very-low-density lipoprotein.

**Figure 3 cancers-15-05557-f003:**
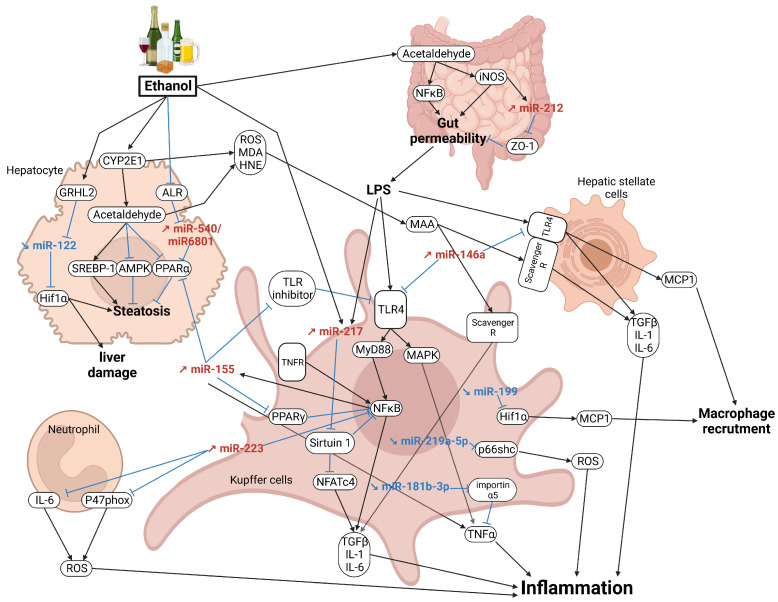
Effect of alcohol consumption on steatohepatitis. Ethanol acts on different cell types (i.e., hepatocytes, Kupffer cells, neutrophils, hepatic stellate cells) in the liver, on intestinal permeability, and on different microRNAs. Black arrows: activation of the pathways. Blue arrows: pathway inhibition. NFκB: nuclear factor kappa B; iNOS: nitric oxidative synthase; ZO-1: Zonula occludens 1; LPS: lipopolysaccharide; CYP2E1: cytochrome P450 2E1; GRHL2: granyhead-like transcription factor 2; Hif1α: hypoxia-inductible factor 1-alpha; SREBP-1: sterol regulatory element binding protein 1; AMPK: AMP-activated protein kinase; PPARα: peroxisome proliferator-activated receptor α; ALR: Augmenter of Liver Regeneration; ROS: reactive oxygen species; MDA: malondialehyde; HNE: 4-hydroxy-2-nonenal; MAA: malondialdehyde-acetaldehyde; TLR4: Toll-like receptor 4; MCP1: monocyte chemotactic protein 1; IL-1: Interleukin-1; IL-6: Interleukin-6; TGFβ: transforming factor β; TNFR: tumor necrosis factor receptor; MyD88: myeloid differentiation response gene 88; MAPK: mitogen-activated protein kinase; PPARγ: peroxisome proliferator-activated receptor γ; NFATc4: nuclear factor of activated T-cells c4; TNFα: tumor necrosis factor α. Created with Biorender.com.

**Figure 4 cancers-15-05557-f004:**
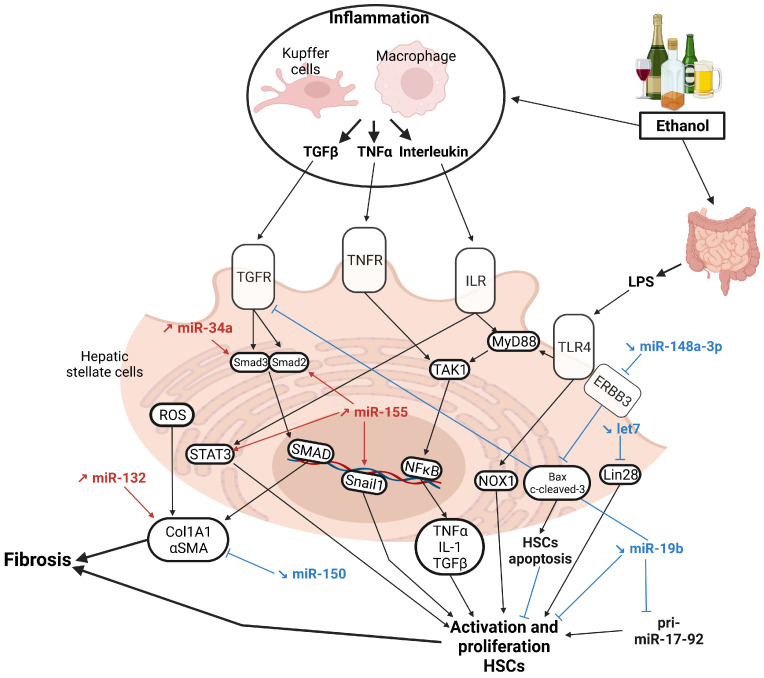
Expression and effects of different deregulated microRNAs inducing hepatic fibrosis in alcoholic cirrhosis. Black and red arrows: activation of the pathways. Blue arrows: pathway inhibition. LPS: lipopolysaccharide; TGFR: transforming factor receptor; TNFR: tumor necrosis factor receptor; ILR: Interleukin receptor; TLR4: Toll-like receptor 4; ERBB3: receptor tyrosine-protein kinase; MyD88: myeloid differentiation response gene 88; NFκB: nuclear factor kappa B; ROS: reactive oxygen species; Col1A1: Collagen 1a1; αSMA: α-Smooth muscle actin; TNFα: tumor necrosis factor α; IL-1: Interleukin-1; TGFβ: transforming factor β; NOX1: NADPH oxidase 1; HSCs: Hepatic stellate cells. Created with Biorender.com.

**Figure 5 cancers-15-05557-f005:**
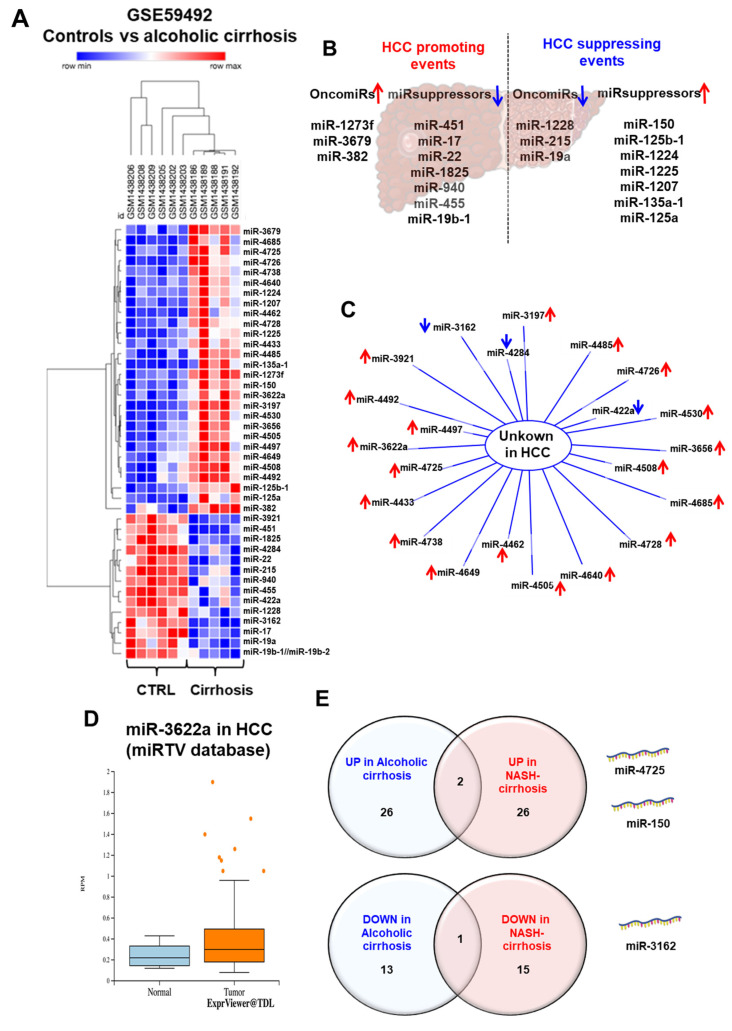
(**A**) A transcriptomic dataset (GSE59492 from Gene Expression Omnibus Database) was used to analyze deregulated miRNAs between control and alcohol-related cirrhotic livers (**B**) A literature-based screening was used to classify them in oncomiRs or tumor suppressor miRNAs (miRsupressors). (**C**) Among deregulated miRNAs, some have unknown functions in HCC. (**D**) Overexpression of some of these microRNAs, such as miR-3622a, can be observed in HCC tumors (data retrieved from miRTV database in July 2023). (**E**) The same transcriptomic dataset (GSE59492) was used to compare deregulated miRNAs in alcohol-related cirrhosis with NASH related cirrhosis. Only two miRNAs (miR-4725 and miR-150) are commonly upregulated in both conditions, and one miRNA is commonly downregulated (miR-3162).

**Figure 6 cancers-15-05557-f006:**
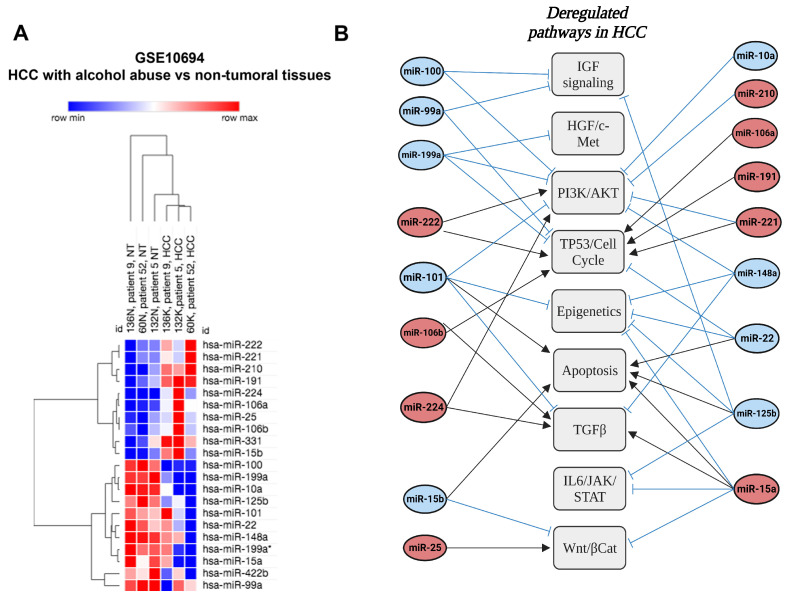
(**A**) A transcriptomic dataset (GSE10694) was used to identify new miRNAs deregulated in hepatocellular carcinoma with alcohol abuse. The data are represented in a heatmap showing the log2 fold change of deregulated miRNAs. (**B**) Significantly deregulated microRNAs were subjected to literature-based screening to classify them in the most common HCC-related pathways. The data were retrieved in July 2023. miRNAs in red bubbles: upregulated; miRNA in blue bubbles: downregulated. Black arrows: activation of the pathway; Blue inhibitory arrows: pathway inhibition. Created with Biorender.com.

**Figure 7 cancers-15-05557-f007:**
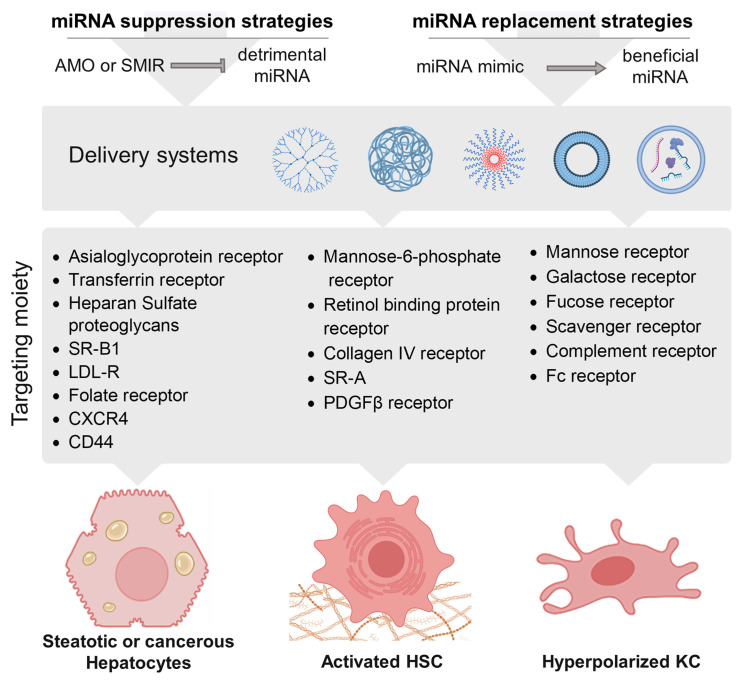
Graphical representation of the miRNA suppression and replacement strategies and list of specific receptors of hepatic cell type used for associating targeting moiety to delivery systems. AMO: anti-miRNA oligonucleotide; SMIR: small molecule inhibitor of miRNA; SR-B1: scavenger receptor class B type 1; LDL-R: low-density lipoprotein receptor; CXCR4: C-X-C chemokine receptor type 4, SR-A: scavenger receptor type A; PDGFβ: platelet-derived growth factor beta; Fc: fragment crystallizable; HSC: hepatic stellate cell; KC: Kupffer cell.

**Table 3 cancers-15-05557-t003:** Examples of miRNA-based clinical trials which are deregulated in ALD or ALD-related HCC (clinicaltrials.gov, retrieved in 25 October 2023).

Identification	Title	Phase	miRNA Target	Disease
**NCT01727934**	Miravirsen Study in Null Responder to Pegylated Interferon Alpha Plus Ribavirin Subjects with Chronic Hepatitis C	II	miR-122	Hepatitis C virus infection
**NCT02862145**	Pharmacodynamics Study of MRX34, MicroRNA Liposomal Injection in Melanoma Patients with Biopsy Accessible Lesions (MRX34-102)	I	miR-34	Advanced melanoma
**NCT03373786**	A Study of RG-012 in Subjects with Alport Syndrome	I	miR-21	Alport syndrome
**NCT02369198**	MesomiR 1: A Phase I Study of TargomiRs as 2nd or 3rd Line Treatment for Patients with Recurrent MPM and NSCLC	I	miR-16	Malignant Pleural Mesothelioma (MPM) and Advanced Non-Small Cell Lung Cancer (NSCLC)
**NCT03601052**	Efficacy, Safety, and Tolerability of Remlarsen (MRG-201) Following Intradermal Injection in Subjects with a History of Keloids	II	miR-29	Keloid formation
**NCT03837457**	PRISM: Efficacy and Safety of Cobomarsen (MRG-106) in Subjects with Mycosis Fungoides Who Have Completed the SOLAR Study (PRISM)	II	miR-155	Cutaneous T-Cell Lymphoma (CTCL) and Mycosis Fungoides (MF)
**NCT0280552**	Safety, Tolerability and Pharmacokinetics of MRG-106 in Patients with Mycosis Fungoides (MF), CLL, DLBCL or ATLL	I	miR-155	Cutaneous T-Cell Lymphoma (CTCL), Mycosis Fungoides (MF), Chronic Lymphocytic Leukemia (CLL), Diffuse Large B-Cell Lymphoma (DLBCL) and Adult T-Cell Leukemia/Lymphoma (ATLL)
**NCT03713320**	SOLAR: Efficacy and Safety of Cobomarsen (MRG-106) vs. Active Comparator in Subjects with Mycosis Fungoides (SOLAR)	II	miR-155	Cutaneous T-Cell Lymphoma (CTCL) and Mycosis Fungoides (MF)
**NCT03603431**	Safety, Tolerability, Pharmacokinetics, and Pharmacodynamics of MRG-110 Following Intradermal Injection in Healthy Volunteers	I	miR-92a	Ischemia

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
