# Peer review of "MiRNAs in Alcohol-Related Liver Diseases and Hepatocellular Carcinoma: A Step toward New Therapeutic Approaches?"

_cancers, 2023, doi:10.3390/cancers15235557_

Round 1

Reviewer 1 Report

Comments and Suggestions for Authors

This paper is interesting, but some points need to be revised:

- Lines 63-37: It is not clear what is the purpose of this review. Can the authors improve this point?

- Lines 263-268: "The PPARα signaling importantly contribute to hepatic inflammation by controlling the NFκB pathway. PPARα inhibits this pro-inflammatory pathway and the expression of associated pro-inflammatory cytokines " What do this paper want to add new about this point?

- It is important to show the role of miRNA in all systems. These 2 important papers should be considered: -- DOI: 10.3390/diagnostics13182888  -- DOI: 10.1016/j.abb.2023.109791  --  DOI: 10.3390/jpm12111925  -- DOI: 10.1016/j.phymed.2023.155091

- Lines 659-662: "Surprisingly, recent findings have demonstrated that the loss of miR-21 in hepatocytes in vivo promotes hepatic carcinogenesis in a model of diethylnitrosamine-treated mice [221]. " improve this point. What do the authors want to say?

- Lines 820-825: "miRNAs-based therapeutics in clinical practices" - clarify the title better

-Lines 875-878:  Move figure 7 higher in the text.

- Lines 868-874: "To conclude, no miRNA targeted delivery system for the treatment of ALD is...  together with a passive-to-active targeting, pave the way for efficient future treatments of ALD." Perhaps this part can be placed in a Future Perspectives section. Revise it.

Comments on the Quality of English Language

Minor editing of English language required

Author Response

This paper is interesting, but some points need to be revised:

We thank the reviewer for the constructive comments, which help us to improve our manuscript.

- Lines 63-37: It is not clear what is the purpose of this review. Can the authors improve this point?

We apologize for the lack of clarity in this paragraph. We have modified this paragraph, as follows:

“The purpose of this review is to discuss the role of microRNAs in the different stages of ALD and how they contribute to the progression toward HCC (HCC priming events). Finally, we discuss the potential strategies that could be employed to target miRNAs in ALD and ALD-related HCC.”

- Lines 263-268: "The PPARα signaling importantly contribute to hepatic inflammation by controlling the NFκB pathway. PPARα inhibits this pro-inflammatory pathway and the expression of associated pro-inflammatory cytokines " What do this paper want to add new about this point?

The goal of this sentence was to remind the importance of PPARα signaling in inflammation before discussing miRNAs targeting PPARα and involved in ASH. For a better clarity, we have modified this part as follows:

“The PPARα signaling importantly contributes to hepatic inflammation by inhibiting the NFκB pathway and the expression of associated pro-inflammatory cytokines [155]. In contrary, PPARγ promotes hepatic steatosis and reduces inflammation [156,157]. Therefore, miRNAs targeting PPARα or PPARγ are likely contributing to the development of ASH.”

- It is important to show the role of miRNA in all systems. These 2 important papers should be considered: -- DOI: 10.3390/diagnostics13182888 -- DOI: 10.1016/j.abb.2023.109791  --  DOI: 10.3390/jpm12111925  -- DOI: 10.1016/j.phymed.2023.155091

These references, DOI: 10.3390/diagnostics13182888 and DOI: 10.1016/j.abb.2023.109791 and DOI: 10.1016/j.phymed.2023.155091, have been added in the paragraph “MicroRNAs in Alcoholic Hepatitis (AH)/circulating miRNAs:

“Secreted miRNAs importantly contribute to intercellular crosstalk [211,212] and the regulation of several physiological and pathological processes [213,214] (PMID: 37858665, PMID: 37844378), including inflammation [154]. Moreover, circulating miRNAs can be detected in body fluids and thus may represent novel biomarkers for a wide range of human diseases [215] (PMID: 37761255).”

 The reference DOI: 10.3390/jpm12111925, has been added in the introduction:

“Accordingly, alteration of miRNAs expression or activity contributes to the development of several diseases [24-27] (36422101, 37269474, 18817506, 37108432).”

- Lines 659-662: "Surprisingly, recent findings have demonstrated that the loss of miR-21 in hepatocytes in vivo promotes hepatic carcinogenesis in a model of diethylnitrosamine-treated mice [221]. " Improve this point. What do the authors want to say?

We apologize for the lack of clarity. Herein, our goal was to suggest cautions regarding miR-21’functions in HCC. Indeed, although some studies indicate its overexpression in HCC and tumor promoting functions, its loss in vivo, in the diethylnitrosamine model of HCC, promotes tumor development and thus suggest tumor suppressive properties. We have modified this sentence for a better clarity:

“Surprisingly, recent findings have demonstrated that the loss of miR-21 in hepatocytes in vivo promotes hepatic carcinogenesis in a model of diethylnitrosamine-treated mice [238], thus suggesting that miR-21 can also exert tumor suppressive properties.”

- Lines 820-825: "miRNAs-based therapeutics in clinical practices" - clarify the title better

We have modified the title as follows:

“Therapeutic approaches targeting miRNAs in clinical trials and future perspectives “

-Lines 875-878:  Move figure 7 higher in the text.

We moved the figure higher in the text.

- Lines 868-874: "To conclude, no miRNA targeted delivery system for the treatment of ALD is...  together with a passive-to-active targeting, pave the way for efficient future treatments of ALD." Perhaps this part can be placed in a Future Perspectives section. Revise it.

We agree with the reviewer, and we have moved this paragraph in the conclusion.

“Although ALD is the most prevalent liver disease in developed countries, they are currently no reviews documenting the role of miRNAs in the all the stages of this disease. Our study is not only providing an exhaustive overview of the role of miRNAs in the development of ALD but also provides evidence that deregulated miRNAs at each stage of the disease contribute to the establishment of a neoplastic phenotype. More than one hundred miRNAs are discussed, thus highlighting the importance of post-transcriptional regulation of gene expression in ALD and HCC and raising many questions regarding the therapeutic targeting of these miRNAs. Currently, they are no miRNA targeted delivery system for the treatment of ALD on the market. Although many strategies can be designed to efficiently target these miRNAs, it remains to determine which ones should be targeted. Moreover, more suitable in vivo models are tremendously required to characterize the role of these miRNAs in ALD/HCC and evaluate the potential of their therapeutic potential. The very first stages of ALD, including steatosis, are not a primary source of research and the targeted delivery of miRNA mainly focuses on the later stages like fibrosis resolution or HCC remission. The development of dual therapeutics, combining several drugs (anti-miR and anticancer) or targeting several cell types (KC and HSC), together with a passive-to-active targeting, pave the way for efficient future treatments of ALD. What's more, increasing evidence challenge the dogmatic view of miRNAs as strict inhibitor of gene expression, and suggest in contrary that miRNAs can induce gene expression [336]. Finally, it should be reminded that miRNA-dependent regulation is a complex process tightly regulated by other trans-acting factors (e.g., lncRNAs or RBPs), which regulate the bioavailability and the activity of miRNAs. Emerging evidence indicate that this interplay is relevant in ALD, as evidenced by mir-214, which is sponged and inactivated by the ethanol-induced lncRNA urothelial cancer-associated 1 (UCA1) in a hepatocyte cell line [337]. The complexity of miRNA-dependent functions is further enhanced by miRNAs editing by specific enzymes (e.g., Adenosine Deaminase, RNA specific, ADAR) controlling miRNA functions and whose expression is often imbalanced in pathological states (i.e., HCC) [338,339].”

Reviewer 2 Report

Comments and Suggestions for Authors

The manuscript is interesting and quite well written. I have some suggestions:

1- Abstract. In this review, we discuss the current knowledge about miRNA’s func-tions in the different stage of ALD and their role in the progression toward carcinogenesis. Finally, we discuss the therapeutic potential of targeting miRNAs for the treatment of these diseases. It might be beneficial to include a sentence in the abstract that briefly summarizes the final key findings of the study. This can provide readers with a quick overview of the research.

2- 1. Introduction L31-34 Fatty Liver Diseases encompasses a spectrum of liver alterations associated with  viral infection (e.g., hepatitis C), obesity, type-2-diabetes (Non-Alcoholic Fatty Liver 33 disease) and chronic/abusive alcohol consumption (Alcoholic Liver Disease) [1]. FLD  starts with the development of hepatic steatosis, where hepatocytes accumulate lipids  (i.e., triglycerides and cholesterol esters) [2]. Please, add some references to better support these sentences.

3- 1. Introduction L64-66. Herein, we are discussing the role of miRNAs in the different stages of ALD, how they can prime the  liver for carcinogenesis, as well as the potential therapeutic approaches that could be  used to target them. Please, improve the description of the aim of the study and underline the novelty of the study.

4- L72-74. Since their discovery in 1993 (lin4 in C-Elegans), more than 38 589 miRNAs (miRBase) have been identified in different organisms (i.e., plants, animals, viruses) among which many have been associated to a wide range of physiological and pathological processes. Please support this sentence with some references.

5- Figure 1. Spectrum of Alcohol-related liver disease with deregulated microRNAs at each stage and cited in this review and detailed in Table 1. Percentages represent the rate of patients moving from one stage to another [23]. MicroRNAs in red have increased expression and microRNAs in blue 106 have decreased expression. Created by Biorender.com. Please, improve this figure.

6- Figure 2. This is an interesting figure, please improve the quality.

7- 9. Conclusion L882-883. In this manuscript, we provide an exhaustive overview of the role of miRNAs in the different stages of ALD and how they can prime the liver for hepatic carcinogenesis. Please explain the acronyms at the beginning of each new paragraph. Underline in the conclusions the novelty of the study and the possible clinical consequences.

Comments on the Quality of English Language

 Minor changes of English language are required

Author Response

The manuscript is interesting and quite well written. I have some suggestions:

1- Abstract. In this review, we discuss the current knowledge about miRNA’s functions in the different stage of ALD and their role in the progression toward carcinogenesis. Finally, we discuss the therapeutic potential of targeting miRNAs for the treatment of these diseases. It might be beneficial to include a sentence in the abstract that briefly summarizes the final key findings of the study. This can provide readers with a quick overview of the research.

We have modified the abstract:

“In this review, we discuss the current knowledge about miRNA’s functions in the different stages of ALD and their role in the progression toward carcinogenesis. We highlight that each stage of ALD is associated with deregulated miRNAs involved in hepatic carcinogenesis and thus represent HCC-priming miRNAs. By using in silico approaches, we have uncovered new miRNAs potentially involved in HCC. Finally, we discuss the therapeutic potential of targeting miRNAs for the treatment of these diseases.

2- 1. Introduction L31-34 Fatty Liver Diseases encompasses a spectrum of liver alterations associated with viral infection (e.g., hepatitis C), obesity, type-2-diabetes (Non-Alcoholic Fatty Liver 33 disease) and chronic/abusive alcohol consumption (Alcoholic Liver Disease) [1]. FLD starts with the development of hepatic steatosis, where hepatocytes accumulate lipids (i.e., triglycerides and cholesterol esters) [2]. Please, add some references to better support these sentences.

Three additional references have been added to support our sentences:

  1. Devarbhavi, H.; Asrani, S.K.; Arab, J.P.; Nartey, Y.A.; Pose, E.; Kamath, P.S. Global Burden of Liver Disease: 2023 Update. J. Hepatol. 2023, 79, 516–537, doi:10.1016/j.jhep.2023.03.017.
  2. Macpherson, I.; Abeysekera, K.W.M.; Harris, R.; Mansour, D.; McPherson, S.; Rowe, I.; Rosenberg, W.; Dillon, J.F.; Yeoman, A.; Specialist Interest Group in the Early Detection of Liver Disease Members Identification of Liver Disease: Why and How. Frontline Gastroenterol. 2022, 13, 367–373, doi:10.1136/flgastro-2021-101833.
  3. Idilman, I.S.; Ozdeniz, I.; Karcaaltincaba, M. Hepatic Steatosis: Etiology, Patterns, and Quantification. Semin. Ultrasound CT MRI 2016, 37, 501–510, doi:10.1053/j.sult.2016.08.003.

“Fatty Liver Diseases encompasses a spectrum of liver alterations associated with viral infection (e.g., hepatitis C), obesity, type-2-diabetes (Non-Alcoholic Fatty Liver disease) and chronic/abusive alcohol consumption (Alcoholic Liver Disease) [1–3]. FLD starts with the development of hepatic steatosis, where hepatocytes accumulate lipids (i.e., triglycerides and cholesterol esters) [4,5]. ”

3- 1. Introduction L64-66. Herein, we are discussing the role of miRNAs in the different stages of ALD, how they can prime the liver for carcinogenesis, as well as the potential therapeutic approaches that could be used to target them. Please, improve the description of the aim of the study and underline the novelty of the study.

We have modified the end of the introduction to better explain the goal of our study (line 68-73):

“Although intense efforts have been devoted to characterizing miRNA’s functions in the context of NAFLD [28,29], limited amount of knowledge are available for ALD and ALD-associated HCC. The purpose of this review is to discuss the role of microRNAs in the different stages of ALD and how they contribute to the progression toward HCC (HCC priming events). Finally, we discuss the different strategies that could be employed to target miRNAs in ALD and ALD-related HCC.”

4- L72-74. Since their discovery in 1993 (lin4 in C-Elegans), more than 38 589 miRNAs (miRBase) have been identified in different organisms (i.e., plants, animals, viruses) among which many have been associated to a wide range of physiological and pathological processes. Please support this sentence with some references.

We have now added new references to support this sentence: “Since their discovery in 1993 (lin4 in C-Elegans) [32] (PMID: 8252621), more than 38 589 miRNAs (miRBase) have been identified in different organisms (i.e., plants, animals, viruses) among which many have been associated to a wide range of physiological and pathological processes [24–27] (PMIDs 36422101, 37269474, 18817506).”

  1. Montemurro, N.; Ricciardi, L.; Scerrati, A.; Ippolito, G.; Lofrese, G.; Trungu, S.; Stoccoro, A. The Potential Role of Dysregulated MiRNAs in Adolescent Idiopathic Scoliosis and 22q11.2 Deletion Syndrome. J. Pers. Med. 2022, 12, 1925, doi:10.3390/jpm12111925.
  2. Chimenti, C.; Magnocavallo, M.; Vetta, G.; Alfarano, M.; Manguso, G.; Ajmone, F.; Ballatore, F.; Costantino, J.; Ciaramella, P.; Severino, P.; et al. The Role of MicroRNA in the Myocarditis: A Small Actor for a Great Role. Curr. Cardiol. Rep. 2023, 25, 641–648, doi:10.1007/s11886-023-01888-5.
  3. Lee, Y.S.; Dutta, A. MicroRNAs in Cancer. Annu. Rev. Pathol. 2009, 4, 199–227, doi:10.1146/annurev.pathol.4.110807.092222.
  4. Pekarek, L.; Torres-Carranza, D.; Fraile-Martinez, O.; García-Montero, C.; Pekarek, T.; Saez, M.A.; Rueda-Correa, F.; Pimentel-Martinez, C.; Guijarro, L.G.; Diaz-Pedrero, R.; et al. An Overview of the Role of MicroRNAs on Carcinogenesis: A Focus on Cell Cycle, Angiogenesis and Metastasis. Int. J. Mol. Sci. 2023, 24, 7268, doi:10.3390/ijms24087268.
  5. Lee, R.C.; Feinbaum, R.L.; Ambros, V. The C. Elegans Heterochronic Gene Lin-4 Encodes Small RNAs with Antisense Complementarity to Lin-14. Cell 1993, 75, 843–854, doi:10.1016/0092-8674(93)90529-y.

5- Figure 1. Spectrum of Alcohol-related liver disease with deregulated microRNAs at each stage and cited in this review and detailed in Table 1. Percentages represent the rate of patients moving from one stage to another [23]. MicroRNAs in red have increased expression and microRNAs in blue 106 have decreased expression. Created by Biorender.com. Please, improve this figure.

We have increased the size of the legends in the figure for a better clarity.

The size of the figure has also been increased.

6- Figure 2. This is an interesting figure, please improve the quality.

The size of the legend has been increased, as well as the thickness of the arrows. The resolution is high enough for publication.

7- 9. Conclusion L882-883. In this manuscript, we provide an exhaustive overview of the role of miRNAs in the different stages of ALD and how they can prime the liver for hepatic carcinogenesis. Please explain the acronyms at the beginning of each new paragraph.

Based on MDPI’s guidelines, acronyms should be explained once and not at the beginning of each new paragraph. We have carefully checked the conclusion to ensure that all acronyms have previously been explained. Only the explanation for UCA1 was missing: “urothelial cancer associated 1”.

Underline in the conclusions the novelty of the study and the possible clinical consequences.

We have modified the conclusion to better highlight the novelty of the study:

“Although ALD is the most prevalent liver disease in developed countries, they are currently no reviews documenting the role of miRNAs in the all the stages of this disease. Our study is not only providing an exhaustive overview of the role of miRNAs in the development of ALD but also provides evidence that deregulated miRNAs at each stage of the disease contribute to the establishment of a neoplastic phenotype. More than one hundred miRNAs are discussed, thus highlighting the importance of post-transcriptional regulation of gene expression in ALD and HCC and raising many questions regarding the therapeutic targeting of these miRNAs. Currently, they are no miRNA targeted delivery system for the treatment of ALD on the market. Although many strategies can be designed to efficiently target these miRNAs, it remains to determine which ones should be targeted. Moreover, more suitable in vivo models are tremendously required to characterize the role of these miRNAs in ALD/HCC and evaluate the potential of their therapeutic potential. The very first stages of ALD, including steatosis, are not a primary source of research and the targeted delivery of miRNA mainly focuses on the later stages like fibrosis resolution or HCC remission. The development of dual therapeutics, combining several drugs (anti-miR and anticancer) or targeting several cell types (KC and HSC), together with a passive-to-active targeting, pave the way for efficient future treatments of ALD. What's more, increasing evidence challenge the dogmatic view of miRNAs as strict inhibitor of gene expression, and suggest in contrary that miRNAs can induce gene expression [336]. Finally, it should be reminded that miRNA-dependent regulation is a complex process tightly regulated by other trans-acting factors (e.g., lncRNAs or RBPs), which regulate the bioavailability and the activity of miRNAs. Emerging evidence indicate that this interplay is relevant in ALD, as evidenced by mir-214, which is sponged and inactivated by the ethanol-induced lncRNA urothelial cancer-associated 1 (UCA1) in a hepatocyte cell line [337]. The complexity of miRNA-dependent functions is further enhanced by miRNAs editing by specific enzymes (e.g., Adenosine Deaminase, RNA specific, ADAR) controlling miRNA functions and whose expression is often imbalanced in pathological states (i.e., HCC) [338,339].”

Reviewer 3 Report

Comments and Suggestions for Authors

This narrative review presented by Jouve et al. manages to combine an interesting and hot topic at the time of current biomedical research. The manuscript needs minor changes before final acceptance:

-The title should not be so generic, it should be more translational.

-Please review the keywords. The authors should be more clinical.

-Clinical trials must be included in a specific section.

-Authors must include the manuscripts doi: 10.3390/ijms24087268. and doi: 10.3390/cancers15184651.

-The authors must improve the use of English grammar.

Comments on the Quality of English Language

Moderate editing of English language required.

Author Response

This narrative review presented by Jouve et al. manages to combine an interesting and hot topic at the time of current biomedical research. The manuscript needs minor changes before final acceptance:

We thank the reviewer for his/her time to review our manuscript and for the constructive comments, which allowed us to improve our review.

-The title should not be so generic; it should be more translational.

We have modified the title as follows:

“MiRNAs in Alcoholic-Related Liver Diseases and Hepatocellular Carcinoma: toward new therapeutic approaches?”

-Please review the keywords. The authors should be more clinical.

Our keywords are fitting well with the fundamental aspects of our manuscript. However, we have added “miRNAs-based therapeutics” to be more clinical, as suggested by the reviewer.

-Clinical trials must be included in a specific section.

Line 828: we have modified the title to highlight the clinical trials. As discussed in our paragraph, miRNA-based therapies, patented, approved, or marketed medicine have been extensively reviewed [290,293,316,317] in the context of HCC or HCV infection. However, they are currently no clinical trials regarding miRNAs targeting in the context of ALD or ALD-related HCC. The goal of this section is therefore to suggest some perspectives to targets miRNAs in the context of alcohol. We have added a small table to document some clinical trials for miRNAs deregulated in ALD or ALD-related HCC.

-Authors must include the manuscripts doi: 10.3390/ijms24087268. and doi: 10.3390/cancers15184651.

The following reference, doi: 10.3390/ijms24087268 (PMID: 37108432), has been added in the introduction and the paragraph 2 (microRNAs). However, the reference doi: 10.3390/cancers15184651 (PMID: 37760618) is not related to ALD, HCC, nor miRNAs. This paper is dealing with the link between IRS4 and oncogenes activation. Because we did not discuss IRS4 in the context of ALD and ALD-related HCC, we believe that this reference is out of the scope of our review.

-The authors must improve the use of English grammar.

All authors have carefully checked the English grammar of our manuscript. We have corrected few errors, which are highlighted in red in the manuscript.

Round 2

Reviewer 1 Report

Comments and Suggestions for Authors

The authors addressed all my criticisms.

Reviewer 2 Report

Comments and Suggestions for Authors

All issues raised in the first review have been addressed

Comments on the Quality of English Language

No further comments